# Stream water sourcing from high elevation snowpack inferred from stable isotopes of water: A novel application of *d-excess* values

Matthias Sprenger[1*], Rosemary W.H. Carroll[2], David Marchetti[3], Carleton Bern[4], Harsh Beria[5], Wendy Brown[6], Alexander Newman[6], Curtis Beutler[6], Kenneth H. Williams[1,6]

[1]Lawrence Berkeley National Laboratory, Berkeley, CA, USA

[2]Desert Research Institute, Reno, NV, USA

[3]Western Colorado University, Gunnison, CO, USA

[4]U.S. Geological Survey, Denver, CO, USA

[5]Department of Environmental Systems Science, ETH Zurich, Zurich, Switzerland

[6]Rocky Mountain Biological Laboratory, Crested Butte, CO, USA

*Corresponding author: msprenger@lbl.gov

**Abstract.** About 80% of the precipitation in the Colorado River's headwaters is snow, and the resulting snowmelt-driven hydrograph is a crucial water source for about 40 million people. Snowmelt from alpine and subalpine snowpack contributes substantially to groundwater recharge and river flow. However, the dynamics of snowmelt progression are not well understood because observations of the high elevation snowpack are difficult due to challenging access in complex mountainous terrain as well as the cost- and labor-intensity of currently available methods. We present a novel approach to infer the processes and dynamics of high elevation snowmelt contributions predicated upon stable hydrogen and oxygen isotope ratios observed in streamflow. We show that deuterium-excess (*d-excess*) values of stream water could serve as a comparatively cost-effective proxy for a catchment integrated signal of high elevation snow melt contributions to catchment runoff.

We sampled stable hydrogen and oxygen isotope ratios of the precipitation, snowpack, and stream water in the East River, a headwater catchment of the Colorado River and the stream water of larger catchments at sites on the Gunnison River and Colorado River.

The *d-excess* of snowpack increased with elevation; the upper subalpine and alpine snowpack (>3200 m) had a substantially higher *d-excess* compared to lower elevations (<3200 m) in the study area. The *d-excess* values of stream water reflected this because *d-excess* values increased as the higher elevation snowpack contributed more to stream water generation later in the snowmelt/runoff season. Endmember mixing analyses based on the *d-excess* data showed that the share of high elevation snowmelt contributions within the snowmelt hydrograph was on average 44% and generally increased during melt period progression, up to 70%. The observed pattern was consistent during six years for the East River, and a similar relation was found for the larger catchments on the Gunnison and Colorado Rivers.

High elevation snowpack contributions were found to be higher for years with lower snowpack and warmer spring
temperatures. Thus, we conclude that the *d-excess* of stream water is a viable proxy to observe changes in high
elevation snowmelt contributions in catchments at various scales. Inter-catchment comparisons and temporal trends
of the *d-excess* of stream water could therefore serve as a catchment-integrated measure to monitor if mountain
systems rely more on high elevation water inputs during snow drought compared to years of average snowpack depths.

## 1    Introduction

The snowpack in mountainous regions provides a crucial water source for the ecosystems and human activities
downstream (Immerzeel et al., 2020). In the alpine and subalpine headwaters of semi-arid regions where the summer
precipitation contribution to streamflow is usually relatively low, as in the southwestern United States, snowmelt
sustains streamflow during much of the growing season when water demands are higher. The Colorado River plays a
special role in the hydrology of the southwestern United States because its headwaters in the Rocky Mountains support
the water supply for about 40 million people, agriculture, industry and power generation (Bureau of Reclamation,
2012). The snowmelt from high elevation upper subalpine and alpine regions of the mountainous headwaters of the
Colorado River was shown to be particularly important for groundwater recharge and sustaining river flow (Carroll et
al., 2019). However, observed (Faybishenko et al., 2022; Hoerling et al., 2019) and projected (Bennett and Talsma,
2021) increases in air temperatures in the headwaters of the Colorado River can lead to a decrease of the snow-to-rain
ratio during the coming decades (Hammond et al., 2023). Therefore, the mountainous catchments in the Colorado
River could likely transition towards low-to-no snow conditions during the second half of this century (Siirila-
Woodburn et al., 2021). In fact, a general trend towards lower snow packs and earlier snowmelt in the western United
States is already observed (Musselman et al., 2021). However, the tools needed to observe high elevation snowmelt
processes are either missing (e.g. point observations), too coarse a resolution (e.g. satellite), or expensive to obtain
(e.g. airborne lidar (Light Detection and Ranging) techniques, numerical models), which is why we investigate the
use of a stable isotope-based method that can help assess upper subalpine and alpine snowmelt contributions to
streamflow.
Snowpack assessments and snowmelt dynamics are usually monitored with point observations like the U.S. Natural
Resource Conservation Service's (NRCS) SNOw TELemetry (SNOTEL) network (NWCC, 2023). However, the
highest elevations in the western United States are not covered by this network (max. elevation 3543 m a.s.l.), despite
this area harboring the largest snow water equivalent (SWE) and most surface water input volumes per square meter
(Hammond et al., 2023). Therefore, although the measured snow pack at SNOTEL sites will indicate melt-out, there
remains substantial snow cover in the alpine regions past the SNOTEL indicated melt-out dates (Dozier et al., 2016).
To obtain a spatial representation of the SWE from the SNOTEL point measurements, regression analyses with
physiographic variables (e.g., elevation, slope, aspect) are commonly used (Fassnacht et al., 2003). Heterogeneity of
snowfall accumulation and redistribution of snow (Freudiger et al., 2017) in complex mountainous terrain makes such
interpolation and extrapolation efforts difficult (Dozier et al., 2016). Adding information about the previous year's
snow cover distribution from satellite data was shown to improve the reconstruction of SWE across the complex
mountainous terrain of the Upper Colorado River Basin (Schneider and Molotch, 2016). However, maps of snowpack
distribution from airborne snow observatory (ASO) based on airborne lidar (Painter et al., 2016) are costly and
therefore may not be applicable across multiple mountainous catchments and/or during several years.
In addition to the high costs and labor intensity of the currently available methods to study high elevation snowmelt
dynamics, these approaches are generally limited to hydrometric data and do not include any tracer information. Beria
et al. (2018) outlined multiple ways how stable hydrogen and oxygen isotopes of water ($\delta^2$H and $\delta^{18}$O) can provide
valuable insights into snow hydrological processes. Because hydrogen and oxygen isotopes comprise the water
molecule, $\delta^2$H and $\delta^{18}$O signatures are ideal tracers to track fluxes in the water cycle (Kendall and McDonnell, 1998).
Stable hydrogen and oxygen isotopes of water have long been used to infer snowmelt contributions to stream water
(e.g., Rodhe, 1981). However, because groundwater recharge is predominantly by snowmelt in snow dominated semi-
arid environments (Sprenger et al., 2022), the isotopic difference between snowmelt newly contributing to the stream
discharge and the groundwater dominated stream flow during baseflow makes mixing model applications unfeasible
in such environments. We therefore explore the applicability of the *d-excess* value as an alternative tracer. This metric
is based on the relation between the hydrogen and oxygen isotope ratios of water systems, which was identified by
Craig (1961a) as
$$\delta^2 H = 8 \times \delta^{18} O + 10 \qquad\qquad (1)$$
and who characterized this relation as indicative of "waters which have not undergone excessive evapotranspiration."
Dansgaard (1964) defined the concept of deuterium-excess, or *d-excess*, as
$$d\text{-}excess = \delta^2 H - 8 \times \delta^{18} O \qquad\qquad (2)$$
which can be interpreted as an index of non-equilibrium in the simple condensation - evaporation of global
precipitation. This formulation has been useful for screening isotopic results from water samples: values of *d-excess*
between 10 and 11 are effectively the intercept in Craig's proposed relation and indicate quasi-stable conditions at a
relative humidity of ~85% (Dansgaard, 1964; Gat, 2000). Here, we test two hypotheses to examine how *d-excess* data
from stream water samples are related to high elevation snowmelt contributions to the catchment runoff during the
snowmelt periods. First, we hypothesize that *d-excess* values in stream water during the snowmelt hydrograph reflect
the changing dominance of snowmelt contributions through time from lower to higher elevations. Second, we test if
these patterns of *d-excess* of stream water are detectable across ranges in drainage area, thus increasing their broader
applicability.

**2    Methods**
**2.1    Study sites and data**
Our study is situated in the headwaters region of the Upper Colorado River (Figure 1) with a focus on an East River
subcatchment (85 km$^2$) as defined by the gaging and sampling station at the Pumphouse location (38.922447, -
106.950828) near Mount Crested Butte, CO. The Pumphouse subcatchment has a large elevation gradient from about
2700 to 4100 m (Figure 1) and is predominantly underlain by Paleozoic and Mesozoic sedimentary rocks, including
Mancos Shale that covers 44% of the catchment area, and localized intrusive igneous rocks like granodiorite (Gaskill
et al., 1991). Varying dominance of vegetation with elevation define four ecozones in the catchment: shrubs, grasses,
and forbs dominate the montane (<2800 m elevation, 2% of catchment area) zone, aspen and conifers dominate in the
lower subalpine (2800 to 3200 m, 34% of the catchment area) region, and conifers dominate in the upper subalpine
(3200 to 3500 m, 32% of the catchment area) region. In the alpine region (>3500 m, 31% of the catchment area),
shrubs are dominant until 3800 m, above which land is mostly barren (Carroll, Deems, Sprenger, et al., 2022).
Meadows are distributed across the catchment but take up a relatively small share of the total area above the montane.
The climate is dominated by cold winters with substantial snow cover and snowpack accumulation that constitutes
about 80% of the total annual precipitation (Carroll, Deems, Sprenger, et al., 2022). There is a consistent snowpack
cover in the subalpine and alpine region with no mid-winter melt. In the montane region melt is very limited (<10
mm/day) prior to early March (Carroll et al., 2022a). The dominant moisture source of winter precipitation in the study
region is the northeastern Pacific and snowfall occurs predominantly from northwestern frontal storms (Marchetti and
Marchetti, 2019). Summers are relatively warm and dry with monsoonal rain that accounts for 20% of the annual
precipitation. The snowpack depth is generally greater and snowmelt timing is later with increasing elevation across
the catchment (Carroll et al., 2022a). The catchment hydrograph is dominated by the snowmelt pulse with an onset in
April, a pronounced peak during June and a subsequent snowmelt recession interspersed with smaller peaks driven by
monsoon rainfall events. Between September and March, the catchment streamflow is generally limited to base flow
(Carroll et al., 2020). The East River has been intensely instrumented and studied since 2015; more details are provided
in Hubbard et al. (2018).
In addition to the East River, we also sampled the Gunnison River near Gunnison, CO, about 50 km downstream from
Mount Crested Butte. This catchment is defined by the USGS streamgage #09114500 (38.54193567, -106.9497661)
and has a drainage area of 2,618 km$^2$. A third basin was included, which is defined by the USGS streamgage #
09095500 (39.2391463 -108.2661946) of the main stem of the Colorado River near Cameo, CO. Its drainage area is
of 20,683 km$^2$ (USGS, 2023). Hereafter, these two basins locations are referred to as Gunnison and Cameo,
respectively, and their catchment areas are shown in Figure 1.
Within the Gunnison River Basin, there are 15 SNOTEL sites located at elevations ranging between 2674 and 3523
m providing snow water equivalent (SWE) observations (Suppl. Table 1). Across these SNOTEL sites, elevation was
not a good predictor for the maximum snowpack depth (Suppl. Fig. 1). For the Colorado River at Cameo, we chose
the 31 SNOTEL sites in the Colorado Headwaters ranging between 2610 and 3452 m (NWCC, 2023) (Figure 1).
We sampled snowpack between 2016 and 2019 across a gradient spanning 1324 m in elevation (from 2347 to 3671
m) in the Gunnison catchment (Figure 1a&b). The snowpack sampling generally took place between early February
and late May with 80% of all samples taken +- 30 days of April 1$^{st}$, which is often assumed to be the timing of peak
SWE. A total of 53 snow pits were dug in flat areas with samples collected in duplicate at 10-cm depth increments to
tabulate snow density, temperature, and stable isotope ratios. Bulk snowpack isotopic content represents the SWE-
weighted composite value across the entire snow column (Carroll et al., 2022b). Precipitation was first sampled on an
event basis via a collector from 2014 to 2017 in Mount Crested Butte at 2885 m ("long-term Precipitation" in Figure
1), and the sampling procedure was outlined in Carroll et al. (2022b). Since 2020, we sampled the precipitation on an
approximate event basis at the locations Estess (2513 m), Mount Crested Butte (2885 m), and Irwin Barn (3181 m)
("Event-based precipitation" in Figure 1). We sampled stream water from the East River at the Pumphouse location
from 2014 to 2022 on daily to fortnightly frequency ("Pumphouse in Figure 1). There was a gap of sampling in April
2018; and therefore, 2018 was excluded from the present analyses. The East River stable isotope data are published
in Williams (2023). Sampling at the Gunnison River was done between March 2020 and December 2021 on a weekly
basis with occasional higher (3 days) or lower (15 days) frequency. At Cameo, stream water sampling occurred at
weekly to fortnightly frequency in 2021 and 2022.
All water samples were measured for stable hydrogen and oxygen isotopes using a Cavity Ring-Down Spectroscopy
(Picarro L2130-i). We report isotope ratios as $\delta^{18}O$ and $\delta^{2}H$ values expressed relative to the Vienna Standard Mean
Ocean Water (Craig, 1961b).

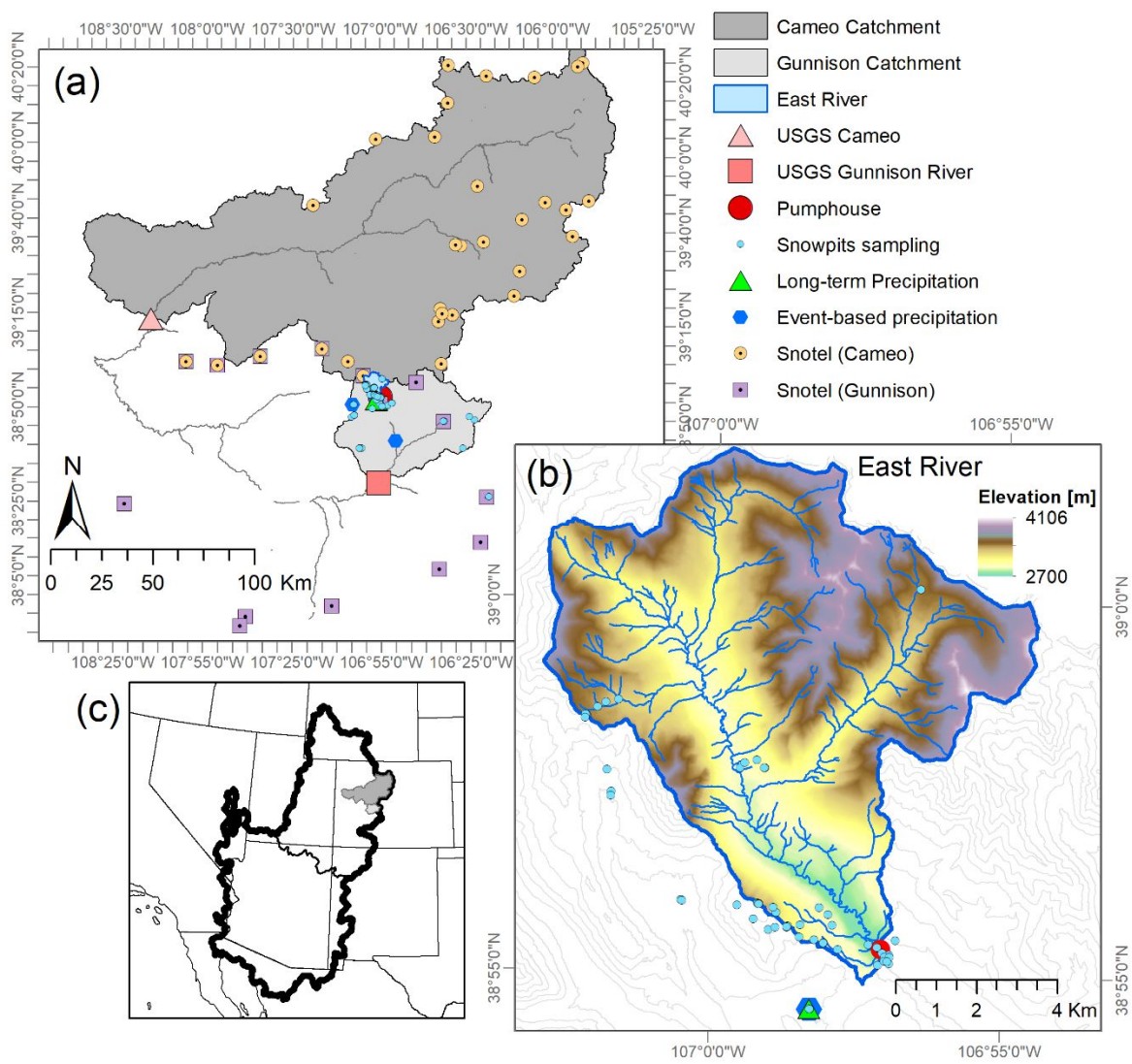


**Figure 1 (a) Locations of streamgages and water sampling of the Colorado River near Cameo and the Gunnison River near Gunnison and the river's catchment area (grey). Locations of event-based precipitation sampling (blue markers), SNOTEL stations in the Colorado River (light blue) and Gunnison River (light purple) areas. East River catchment area (blue outline) as defined by Pumphouse gaging and sampling location (red circle) located within the Gunnison river catchment also shown. (b) Area and elevation of the East River catchment with the streamgage and water sampling location at Pumphouse (red marker) and long-term precipitation sampling site (cyan triangle). (c) Locations of the catchments defined by the stream gages near Cameo and Gunnison (light grey) in the Colorado River Basin (thick black line).**

## 2.2 Data analyses

We calculated the deuterium excess value (short "*d-excess*") for all water samples as defined by equation (2).

The slope of the local meteoric water line is 7.4 (Carroll et al., 2022b) near Mt Crested Butte and 7.2 at the lower elevation Gunnison site (Marchetti and Marchetti, 2019), which does not deviate much from the slope of 8 of the global meteoric water line that defines the *d-excess* (see Suppl. Fig. 2). For significant linear Pearson correlations (p<0.05), we added linear regression lines to the plots.

We used the SNOTEL data to compute the fraction of maximum SWE through time for each water year (a value of
one equals maximum SWE and zero indicates the snowpack is melted). Because SNOTEL SWE data only reflect
conditions at the stations, we used spatially explicit energy balance snowmelt simulations, as published by Carroll et
al. (2022a), that were informed by the spatial variation in SWE as observed by flights of the ASO. For each water year
with snowmelt simulations available, we calculated the cumulative difference through time between the simulated
snowmelt for the montane and alpine elevation bands in the East River, given as millimeter (mm) SWE. In this case,
a value of zero indicated equal snowmelt volumes from the montane and alpine snowpack, whereas positive values
show that alpine snowmelt exceeded montane snowmelt.
We defined the snowmelt period in the East River catchment based on the hydrograph at the Pumphouse streamgage
to be the time between day 200 and 300 of the water year. This period is between Mid-April to late July, because the
water year starts on October 1$^{st}$. For the snowmelt period, we  used the Bayesian mixing model HydroMix, developed
by Beria et al. (2020), to estimate the contribution of high elevation snowmelt to streamflow during the snowmelt
period. HydroMix uses tracer data of the end-members and the mixture to estimate the probability distribution function
(pdf) of the mixing ratio, defined as fractional contribution of end-members to the mixture:
$$\rho S_1 + (1 - \rho)S_2 = M, \tag{3}$$
where $M$ is the tracer concentration in the mixture, $S_1$ and $S_2$ are tracer concentrations in the two sources, and $\rho$ is the
fractional contribution of $S_1$ to mixture $M$.
In typical Bayesian mixing analysis, pdfs are fitted to tracer concentrations in different end-members and the mixture,
and the pdf of the mixing ratio is estimated using standard Bayesian inference principles. This requires a large tracer
dataset to ensure a robust fit to tracers of the end-members and the mixture, which is often not available. HydroMix
adopts a bootstrap approach, using all possible combinations of end-member tracer measurements and formulating a
likelihood function based on an assumed pdf of the underlying error function, which is the difference between
simulated and observed mixture concentration. By using all available combinations of end-member tracer
measurements, HydroMix builds an empirical pdf while optimizing the likelihood function. This approach has been
shown to work both theoretically and in real-case scenarios (Beria et al., 2020).
The two end members ($S_1$ and $S_2$) were defined as the *d-excess* of the snowpack from the upper subalpine and alpine
snowpack (>3200 m, n=31, defined as "high elevation") and lower subalpine and montane area (<3200 m, n=60),
respectively. We report the mean fraction of high elevation snowmelt in each water sample ($M$) with standard
deviations based on the distribution of the two endmembers as described in Beria et al. (2020). We further report the
seasonal flow weighted mean share of high elevation snowpack in the stream samples. We compared the HydroMix
results with MixSIAR (Stock et al., 2018) calculations and found both methods produced very similar results.  Multiple
linear regression was used to explore the predictability of the mean share of high elevation snowmelt during the
different years as a function of the average maximum SWE (SWE$_{Max}$) and the mean air temperature (T$_{air}$) of
measurements at the Gunnison SNOTEL sites (NWCC, 2023) during the snowmelt period.
**3    Results**

### 3.1 The *d-excess* of stream water increased with high elevation snowmelt contributions

Our snowpack sampling campaigns along a 1324 m elevation gradient showed that the average (±SD) *d-excess* value of the high elevation (>3200 m) snowpack was 13.8 (±1.6) ‰ and thus significantly higher than for the lower elevation snowpack 10.7 (±1.8) ‰ (Figure **2**c). The *d-excess* of the lower elevation snowpack was not significantly different from groundwater (10.5±1.0 ‰, Figure **2**c) nor from the *d-excess* of summer rainfall (Suppl. Fig. 3). We further observed a strong and temporally consistent (generally $r > 0.63$ and $p<0.05$ for the four individual years) increase in *d-excess* of the snowpack with elevation (Figure **2**b). The *d-excess* lapse rate of the snowpack was +0.52 ‰/100 m, leading to 12.9 ‰ to 14.4 ‰ and 14.4 ‰ to 17.6 ‰ for the *d-excess* of the snowpack in the upper subalpine and alpine region, respectively. Lapse rates for the snowpack were not seen in $\delta^{18}O$ (Figure **2**b) or $\delta^{2}H$ (data not shown). The precipitation sampled via collectors across the 667 m elevation gradient from the event-based sampler also showed a relation between average *d-excess* and elevation for the samples collected weekly to fortnightly between November and April during water years 2021 and 2022 (Suppl. Fig. 4). These samples reflect a *d-excess* lapse rate for winter precipitation of +0.7 ‰/100 m, which was slightly higher than snowpack, though the elevation range for the precipitation sampler was lower. There was generally a large variability of SWE dynamics across the SNOTEL sites in the Gunnison catchment (Figure 3a), and this variation among the sites did not result from elevation differences (Suppl. Fig. 1).

The hydrograph of the snowmelt period had peak streamflow during May and June, a recession towards August and lowest flows between September and March (Figure 3a). This pattern was consistent during the seven water years, but years with lower SWE resulted in lower peak flows, as expected (Suppl. Fig. 5).

The stream water $\delta^{18}O$ dynamics reflected the seasonality of precipitation inputs, from having lower values (depleted in $^{18}O$) during peak flow and trending towards higher values (enriched in $^{18}O$) during summer and early fall due to greater fractional contributions from base flow and rainfall contributions that had higher $\delta^{18}O$ values compared to the snowfall. Due to the strong difference in $\delta^{18}O$ values of rain and snowfall (see discussion in Sprenger et al., 2022), the $\delta^{18}O$ of stream water decreased during the low flows in winter due to a higher fraction of groundwater sourced from snowmelt vs. rain in the catchment runoff (orange points and line in Figure 3b). The $\delta^{18}O$ of snowmelt stream water reached a minimum in June during maximum snowmelt contribution, after which the snowpack ceased to exist and $\delta^{18}O$ of stream water increased throughout the summer with recession to base flow and monsoonal rainfall.

We found that over the study period, the timing of the peak stream flow could be explained by the timing of the most intense snowmelt (i.e., slope of SWE in Figure 3) and the timing of the complete melt out at the higher (>3200 m) SNOTEL stations (r=0.83 and r=0.79, respectively).

The *d-excess* values of stream water did not show a strong seasonal dynamic, but in general, *d-excess* values mainly increased during the snowmelt season and subsequently dropped again during the summer (red points and line in Figure 3b). The increase of *d-excess* of stream water was not due to rainfall input because there was no seasonal trend in *d-excess* of rainfall (Suppl. Fig. 3). Instead, *d-excess* of stream water resulted from melting snowpack at higher elevations due to snowmelt progression, as evidenced by the SNOTEL SWE data, that resulted in increases in *d-excess*

of stream water consistently for each of the investigated years (Figure 4a). The hypothesis that this increase in *d-*
*excess* of stream water resulted from high elevation snowmelt contributions is supported by its relation with simulated
snowmelt differences between alpine and montane snowmelt volumes through time (Figure 4b). When the high
elevation snowmelt volumes became increasingly larger than the low elevation snowmelt, *d-excess* of stream water
increased consistently. Annual average snowmelt from alpine regions (1075 $m^3$/s) was more than double than
snowmelt from montane regions (520 $m^3$/s), despite the area of the prior (111 $km^2$) being smaller than the latter (143
$km^2$) in Carroll (2022a)'s modeling domain of the East River. Notably, Figure 4b also shows that *d-excess* values of
stream water were highest for years with largest differences between alpine and montane snowpack (2017 and 2019).
Our *d-excess*-based endmember mixing analyses revealed that 41 to 57% of the flow in the East River during the
snowmelt period stemmed from high elevation snowpack (Figure 5). Periods when there were increases in the fraction
of high elevation snowmelt contributions tend to be later in the snowmelt hydrograph and coincided with periods of
runoff intensification (Suppl. Fig. 6). During peak alpine snowmelt contributions, about two-thirds of the East River
flow stemmed from the high elevation snowpack. There was a general trend that the annual mean high elevation
snowpack contributions were higher in water years with lower maximum SWE observed at the SNOTEL sites across
Gunnison county (Suppl. Fig. 7a, r=-0.51, p=0.24). However, the relatively warm snowmelt period of 2017, following
a winter with deep snowpack, resulted in relatively large high elevation snowmelt contributions and thus did not follow
that trend (Suppl. Fig. 7b, r=0.25, p=0.58). Because of this observation, we included in addition to maximum SWE
the average air temperature measured at the SNOTEL sites during the snowmelt period as a second variable in a
multiple regression analysis. The regression equation
*mean high elevation snowmelt contribution = -37.03\*$T_{air}$ -0.73\*$SWE_{max}$ + 0.089\*$T_{air}$\*$SWE_{Max}$ + 350.74*     (4)
explained 66% of the interannual variation of the mean high elevation snowmelt contribution, and all variables had
significance levels of <0.1. Our results therefore indicate that the snowpack at the highest elevation was most important
for runoff generation in low-snow years and relatively high air temperature and years with a deep snowpack and
relatively low air temperature (Figure 6). We also tested the streamflow volumes during the snowmelt period as a
variable, but did not include it, because of its strong correlation with $SWE_{max}$ (r=0.84, p=0.018).


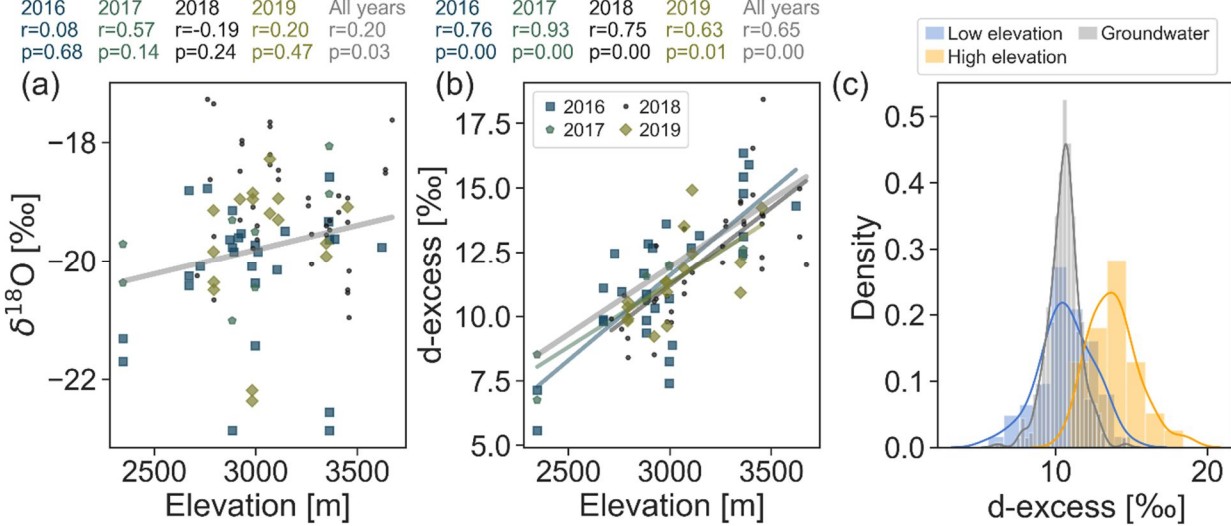


Figure **2** The $\delta^{18}$O (a) and *d-excess* values (b) of the snowpack sampled in the Upper Colorado River Basin during four different winters along an elevation gradient (Carroll et al., 2021). Regression lines are plotted for correlations with p<0.05. For each year and for the bulk isotope data over all years, Pearson correlation coefficients (r) and significant levels (p) are given. (c) Histogram showing the distribution of snowpit *d-excess* values for the sites <3200 m a.s.l. ("Low elevation", blue), sites above >3200 m a.s.l. ("High elevation", orange), and groundwater sampled at five wells between 2015 and 2022 (grey, Williams (2023)). The mean *d-excess* values for the low and high elevation snowpack (10.7 ‰ and 13.8 ‰, respectively) are significantly different (p<0.0001, t = -8.1) according to the t-test. The mean groundwater *d-excess* value (10.5 ‰) is not significantly different from the low elevation snowpack.

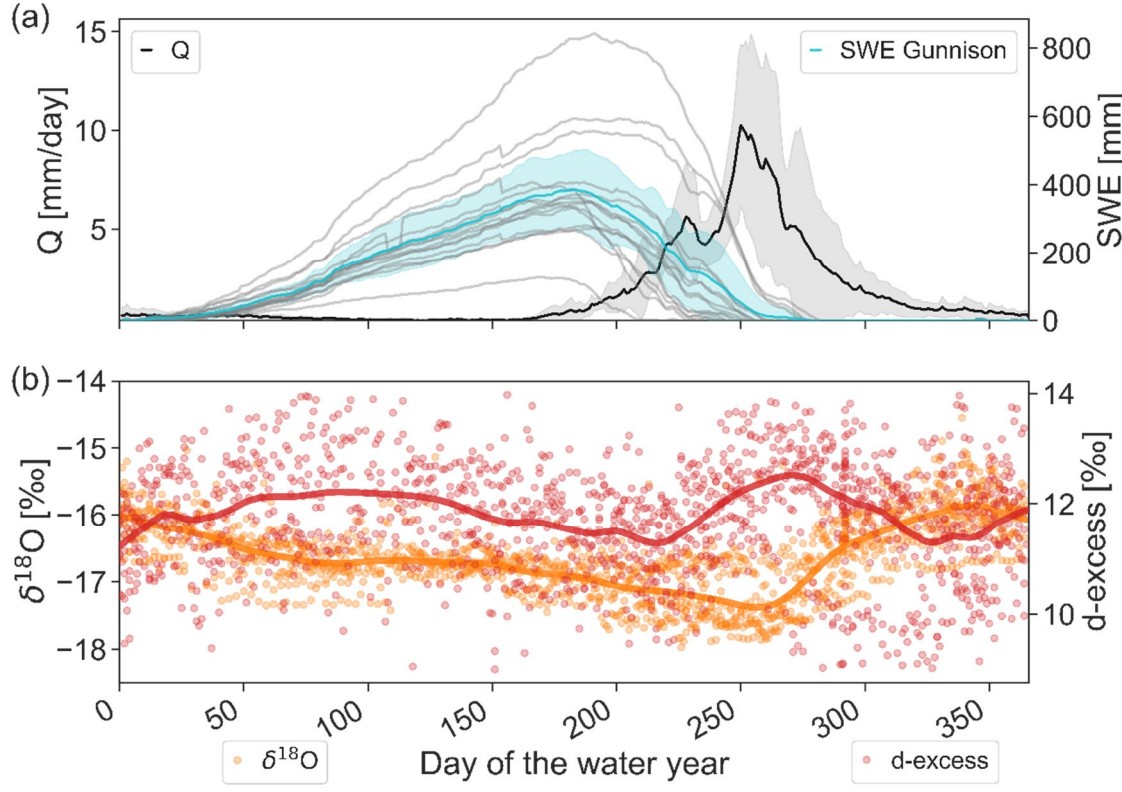

270

**Figure 3 (a) Median annual dynamics of East River streamflow (Q, black, Carroll (2023)) and snow water equivalent (SWE, NWCC (2023)) at the individual SNOTEL sites within the Gunnison River catchment (grey) and the average of all sites (cyan) from water year 2015 to 2022 with semitransparent grey and cyan area representing the standard deviation of Q and SWE, respectively. (b) The $\delta^{18}$O (orange) and *d-excess* (red) of all stream water samples collected between water year 2015 and 2022 from the East River at the Pumphouse location (Williams et al., 2023). The orange and red lines are a LOWESS fit to the data points. See Suppl. Fig. 5 for a time series plot of the same data.**

277

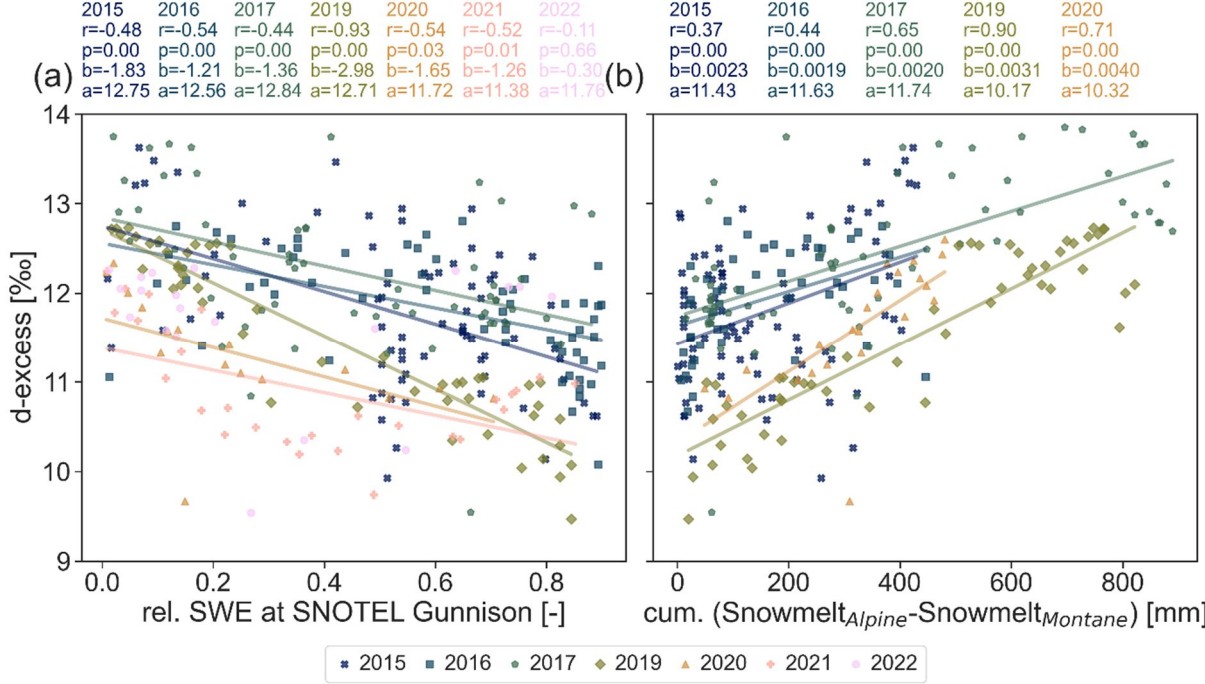

278

**Figure 4 (a) The d-excess of stream water values during snowmelt for seven individual years, shown as a function of relative**
**snow water equivalent (rel. SWE) measured at the SNOTEL stations across the Gunnison River catchment at the time of**
**sampling. For each year, the Pearson correlation (r) and the associated significance level (p) are given as well as the intercept**
**(a) and slope (b) of the regression. (b) The *d-excess* of stream water as a function of the cumulative (cum.) differences**
**between the simulated snowmelt at alpine (=highest elevation in the East River) and montane (lowest elevation in the East**
**River) region at the time of each stream water collection. Regression lines are shown for p≤0.05.**

285

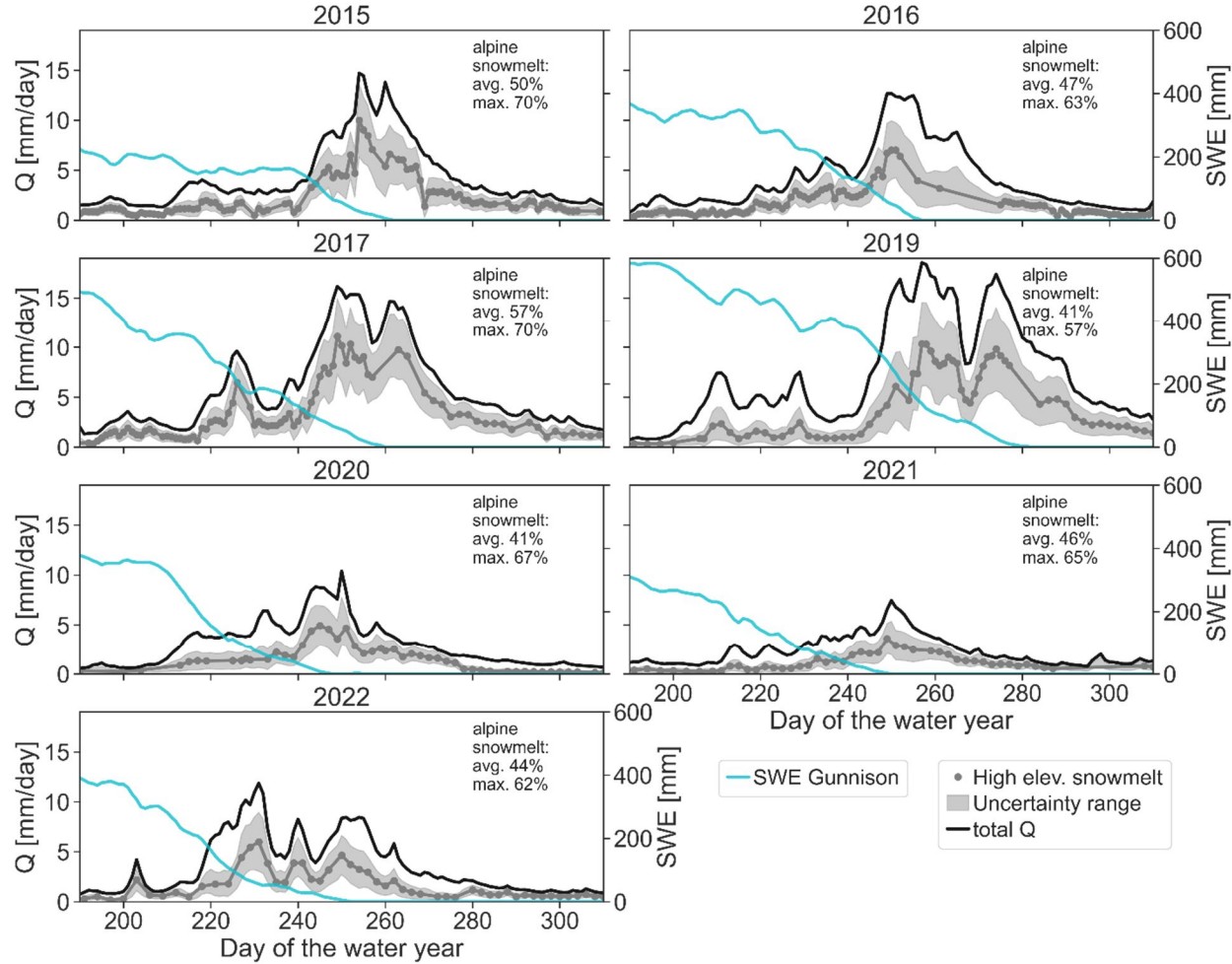

286

**Figure 5 Endmember mixing analyses based on *d-excess* of stream water inferring the share of high elevation snowmelt (grey dots and lines) in the streamflow during the snowmelt-induced peak flow of the East River. The uncertainty range is shown as grey bands and it represents the standard deviation (22% on average). Days 200 and 300 of a water year represent Mid-April and late July, respectively. The cyan line represents the average snow water equivalent (SWE) observed across the SNOTEL sites in Gunnison county.**

292

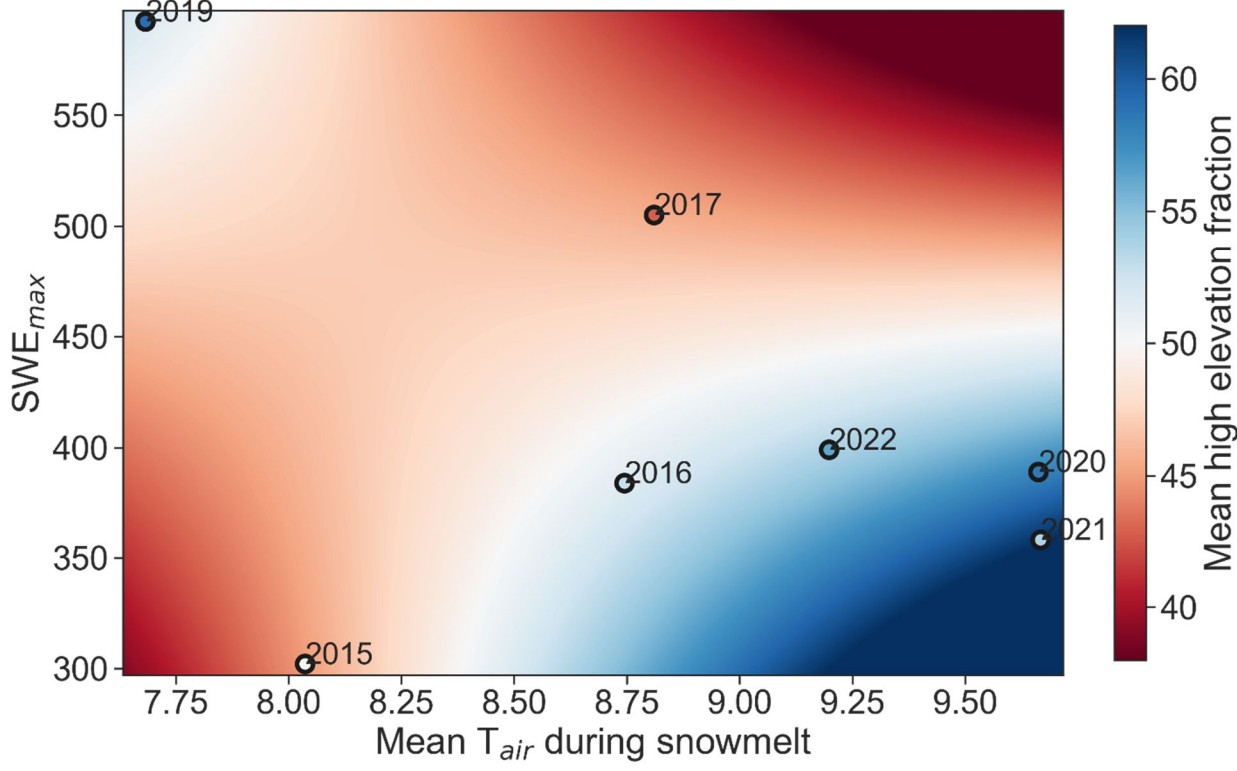

293

**Figure 6 Result of the multiple regression analyses to assess predictability of the mean contribution of high elevation snowmelt to stream water as a function of the maximum snow water equivalent (SWE$_{max}$) and the air temperature (T$_{air}$) during the snowmelt period measured at the SNOTEL sites in Gunnison. Note that the regression includes interaction between SWE$_{max}$ and T$_{air}$.as follows: Maximum high elevation fraction = -37.03\*T$_{air}$ -0.73\*SWE$_{max}$ + 0.089\*T$_{air}$\*SWE$_{Max}$ + 350.74. The data points labelled with years indicate the data that went into the model.**

### 3.2 The *d-excess* dynamics of stream water beyond headwaters

Downstream from the East River, the Gunnison River stream water samples showed similar increase in *d-excess* as streamflow during the snowmelt season increased. This pattern was observed for both years in which stream water sampling in Gunnison was done. In 2020, the snowpack was deeper, and the runoff was higher than in 2021. Additionally, the *d-excess* values of stream water were different for the different years with generally higher values for 2020 than in 2021 (Figure 7a,c). Despite 30 times larger drainage area of the Gunnison River compared to the East River, the effect of the high elevation snowmelt on the *d-excess* measurements of stream water was detectable, albeit dampened given the greater fraction of lower elevations contributing to its flow.

The drainage area of the Colorado River near Cameo is eight times the drainage area of the Gunnison River, but the difference between the *d-excess* of stream water at the beginning and end of the snowmelt period was greater than 3 ‰ in 2021 and 2022. Thus, despite the large catchment area of the Colorado River near Cameo, and greater mixing of runoff in reservoirs within that catchment, the snowmelt contribution from high elevation regions was substantial during the snowmelt peak flow (Figure 7b,d).

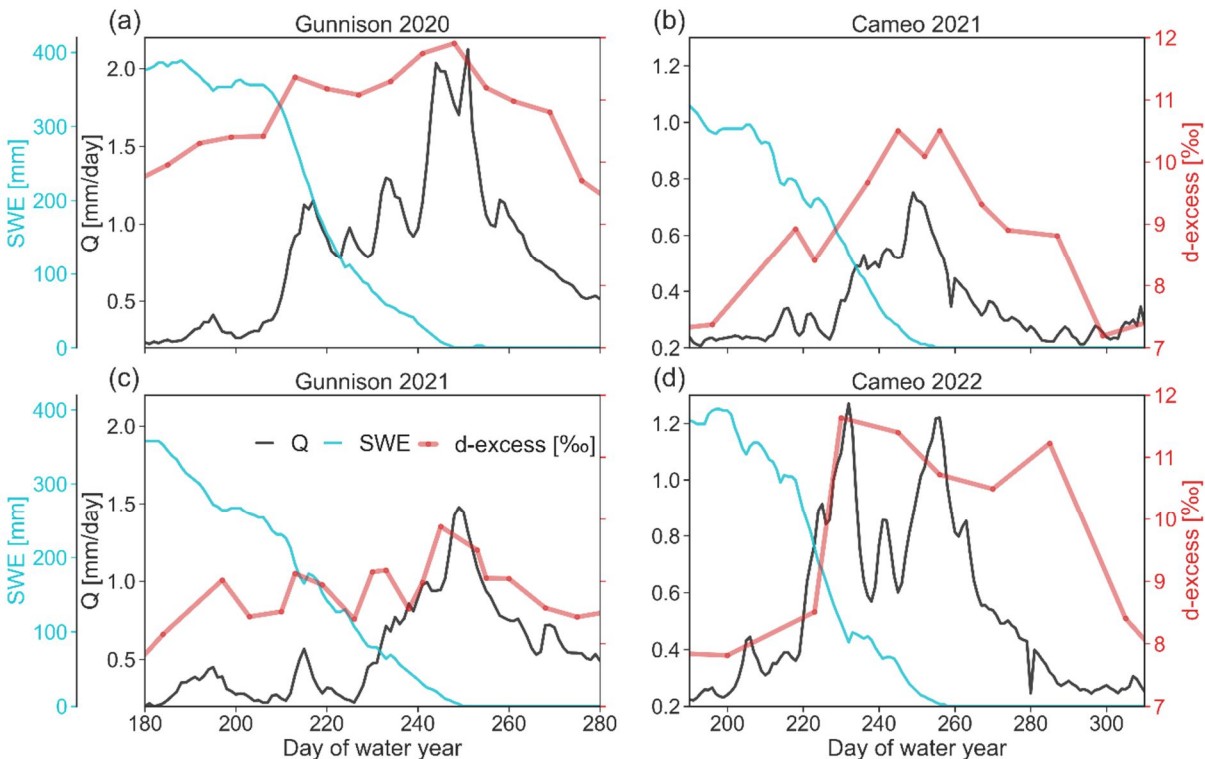

312

**Figure 7 Streamflow (Q, black) and *d-excess* (red dots and line) of the stream water before and during snowmelt for the Gunnison River near Gunnison, Colorado in 2020 (a) and 2021 (c) and for the Colorado River near Cameo, Colorado for 2021 (b) and 2022 (d). Further shown is the average snow water equivalent (SWE, cyan line) of all the SNOTEL sites located in the Gunnison catchment and in the Colorado River headwaters for the Cameo site, respectively. Note that the y-axes have different scales for each subplot.**

## 4    Discussion

### 4.1    The *d-excess* of stream water reflects high elevation snowmelt

We find that *d-excess* of stream water can be used to differentiate the effects of snowmelt from low vs. high elevations using three independent approaches: First, the comparisons of *d-excess* dynamics of stream water with the observed snowpack reduction at SNOTEL sites in the region showed a strong relation that was consistent during six of the seven investigated snowmelt periods (Figure 4a). The SNOTEL data do not show an increased snowpack with elevation (Suppl. Fig. 1), but ASO flight data indicate that snowpack depth generally increases with elevation (Carroll, Deems, Sprenger, et al., 2022). Thus, with decreasing SWE during the snowmelt period, the ratio of high elevation snowmelt can increase. Such a trend of relative increase of the high elevation snowpack during low snow years was observed. Second, simulated differences based on spatially explicit hydrological modeling of snowmelt timing and volumes between the montane and alpine regions within the East River catchment correlated significantly with *d-excess* of stream water for every simulated snowmelt period (Figure 4b). Third, the increase in *d-excess* of stream water coincided with the peak streamflow during each snowmelt period (with exception for 2022, Figure 5). Thus, elevated *d-excess* values cannot stem from low elevation snowmelt but most likely result from higher elevation snowmelt as

the snowmelt generally progresses from lower to higher elevations due to the temperature gradients across the catchment.

Because we observed consistent lapse rates of *d-excess* values in the snowpack during several years (Figure **2**b), significant differences between the *d-excess* at lower and higher elevation snowpack (Figure **2**c), and also a *d-excess* lapse rate in winter precipitation (Suppl. Fig. 4), we see a great potential for *d-excess* measurements to serve as a tracer for endmember-mixing analyses to derive high elevation snowmelt contributions to the catchment's streamflow during snowmelt periods.

Other studies have also shown that winter precipitation (i.e., snow) at highest elevations had the highest *d-excess* values; monthly weighted precipitation data by Froehlich et al. (2008) indicated a lapse rate in *d-excess* values of +0.2 ‰/100 m across an elevation range between 469 and 2245 m in the Alps. Data published by Tappa et al. (2016) indicated a lapse rate of +0.63 ‰/100 m in the Rocky Mountains in Idaho for samples taken between October and May across five sites spanning an elevation gradient from 830 to 1850 m. Rolle (2022) sampled snowpack at ten sites across elevations from 1262 and 1905 m in the Lubrecht Experimental Forest, Greenough, Montana in late March and found a d-excess lapse rate of +0.26 ‰/100 m. Our lapse rate of +0.72 ‰/100 m for precipitation and +0.52 ‰/100 m for the snowpack was higher than in the other studies, but we cover a larger elevation gradient and study higher elevations than the other studies. Nevertheless, the general trend of increased *d-excess* values with elevation was the same for all four studies in mountainous systems.

However, the processes why we see a *d-excess* lapse rate in mountain snowfall and snowpack is not yet fully understood. The current literature suggests two potential processes:

One potential explanation for how *d-excess* lapse rates in the snowpack develop is evaporation and sublimation of snow at lower elevation combined with daytime up-valley (anabatic) winds that occur in mountainous areas and the subsequent condensation of the water vapor at colder higher elevation (Beria et al., 2018; Lambán et al., 2015). Sublimation and evaporation from the snowpack leads to kinetic non-equilibrium fractionation that leaves an isotopically enriched snowpack behind (Stichler et al., 2001). Recent in situ stable isotope measurements by Wahl et al. (2021) support this process, because they saw that when radiation driven sublimation outweighed deposition, the vapor was isotopically depleted compared to the snowpack. They further showed that the isotopic composition of the vapor determined the isotopic composition of the humidity flux during deposition conditions (Wahl et al., 2021). For our study region, we have shown previously via spatially explicit snowmelt modeling based on the energy balance and accounting for isotopic fractionation (Carroll et al., 2022a) that the snowpack at lower elevations experience more snow loss to the atmosphere due to higher energy availability than higher elevation, which lead to an elevation gradient of the *d-excess* in the simulations. These simulations also have shown that shading provided by vegetation in forested areas reduces evaporation and sublimation from the underlying snowpack, making *d-excess* values of these snowpack higher than snowpack in non-forested areas at the same elevation (Carroll et al., 2022a). Because the snowpack in forests with higher *d-excess* values melt later than the snowpack in non-forested areas, it also results in an increase in stream water *d-excess* values during the later phase of the snowmelt discharge peak.

The second potential explanation for how *d-excess* lapse rates in the snowpack develop would be sub-cloud evaporation, which leads to lower *d-excess* values of precipitation at lower elevations, because the distance between

cloud base and ground and the saturation deficit are higher than at higher elevations. Thus, precipitation at lower
elevations would experience more kinetic non-equilibrium isotopic fractionation due to evaporation leading to lower
*d-excess* (Froehlich et al., 2008). However, this process is less like to occur during winter time and snowfall (Froehlich
et al., 2008), and Xing et al. (2023) showed with precipitation and vapor isotope measurements that sub-cloud
evaporation altered the *d-excess* values of snowfall much less than rainfall in the Chinese Loess Plateau. While we
cannot conclude which process leads to the *d-excess* lapse rate, the observation of a *d-excess* lapse rate in several other
high elevation snow studies (Rolle, 2022; Tappa et al., 2016; Froehlich et al., 2008) suggests that we could expect a
*d-excess* response due to high elevation snowmelt contributions in the flow of other mountainous streams. Thus, the
transferability of our approach to other watersheds will depend on observations of a *d-excess* lapse rate in the
snowpack, which will likely be influenced by climatic conditions that lead to thick a snowpack without mid-winter
melt, relatively steady moisture source of the snowfall, and accessibility to sample the snowpack near peak SWE.
Importantly, our long-term sampling of the precipitation in the East River can rule out a potential precipitation *d-*
*excess* seasonality to influence the *d-excess* of stream water during the snowmelt period (Suppl. Fig. 3). Therefore,
there are several independent data sources that all point towards high elevation snowmelt contributions to the
catchment streamflow driving the observed *d-excess* of stream water variation during the snowmelt period.
Our findings, based on endmember-mixing analyses via *d-excess* values highlight the importance of high elevation
snowpack for runoff generation. Since the d-excess values in the groundwater are more similar to the lower elevation
snowpack (Figure **2**c), we infer that groundwater recharge is dominated by early snowmelt in relatively lower
elevations infiltrating into a relatively dry subsurface. High elevation snowmelt occurs during later freshet when the
soils are already saturated or near saturation, which leads to fast runoff generation and thus shorter travel times and
higher runoff efficiency (as outlined by Webb et al., 2022) of high elevation snowmelt than low elevation snowmelt.
This temporal aspect of the high elevation snowmelt and its larger contribution to streamflow later in the snowmelt
hydrograph is reflected in the endmember mixing results that show the highest share on the recession limb of the
hydrograph (Suppl. Fig. 6). The interannual variation in *d-excess* of stream water and the derived high elevation
snowmelt contributions indicate that the snowpack of the upper subalpine and alpine region could be most important
in years of relatively low snowpack accumulation and comparably high spring air temperatures. The observed
regression stems from the generally higher volume share of high elevation snowpack comparted to low elevation
snowpack during low snow years, and the faster melt out during warmer spring temperatures, both leading to larger
contributions of high elevation snowmelt to the spring hydrograph peak. Thus, with the projection of a reduced
snowpack in the western United States (Siirila-Woodburn et al., 2021), understanding the high elevation snowpack
dynamics could most likely become more important, and *d-excess* observations are a tool to investigate the timing
(e.g., trend towards earlier melt) and fate (e.g., streamflow contribution vs. sublimation or groundwater recharge) of
the snowpack throughout the melting period.
**4.2    Limitations and opportunities of *d-excess* of stream water with scale**
Our results show that the *d-excess* patterns of stream water observed in a headwater stream can be upscaled because
we see a similar *d-excess* pattern of stream water at larger scales from stream water sampling at the USGS streamgages
of the Gunnison near Gunnison and Colorado River near Cameo. The latter sampling site is an entirely different
catchment to the north of East River and Gunnison River in which the snowpack was not sampled for its *d-excess*
values. However, the *d-excess* signal of stream water for Coal Creek, a smaller headwater catchment to the west of
the East River catchment, did not show a similar pattern (Suppl. Fig. 8, Suppl. Fig. 9), likely because of a lower
representation of high elevation bands within in the catchment (Suppl. Fig. 10). Twenty nine percent of the Coal Creek
catchment area is the upper subalpine region, but only 6% of the catchment is alpine (>3500 m). Thus, high elevation
snowpack with the highest *d-excess* values is essentially missing in Coal Creek, which presumably dampened *d-excess*
response of stream water. We therefore hypothesize that the applicability of the *d-excess* of stream water as a signal
for high elevation snowmelt is dependent on a sufficient area with high elevation (>3200 m) and sufficient elevation
gradient in the catchment of the sampled stream. Lastly, although we see *d-excess* dynamics of stream water in
response to high elevation snowmelt at relatively large scales, the isotope dynamics may likely not be detectable
downstream from large reservoirs. Initial sampling of the Colorado River near the Colorado-Utah state line with a
drainage area of 46,230 km$^2$ that includes several large reservoirs indicates that stream water *d-excess* changes are
rather dampened and might not hold sufficient information to infer high elevation snowmelt contributions (not shown).
Because snowpack volumes are getting lower, and snowmelt is starting earlier in mountainous regions due to climate
change (Musselman et al., 2021), we could benefit by finding ways to assess the effect of these both at sub-annual to
decadal time scales. Short term identification of a snow drought could allow for adaptive water management measures
on the sub-annual time scale, whereas long-term trends might show the trajectory of mountain snow dynamics. With
0.2 ‰ measurement uncertainty of the *d-excess* values due to 0.025 ‰ and 0.1 ‰ precision (1σ) in $\delta^{18}O$ and $\delta^2H$,
respectively, the observed variation of *d-excess* in snowpack and stream water are at least ten times larger. Our results
and the discussion in the previous section show that measurements of *d-excess* of stream water is a relatively efficient
way to obtain catchment integrated information about the high elevation snowpack.
Although SNOTEL sites are point measurements and therefore do not represent integrated patterns across
heterogeneous mountainous regions, *d-excess* of stream water does integrate throughout catchment areas. The lidar
based ASO data provide spatially explicit snowpack observations on catchment scales, but such data collection can
be difficult and represents only snapshots in time, although time series changes of snowpack during the snowmelt
period might be more informative. The difficulty of large-scale flight-based data collection may also make monitoring
of interannual SWE changes difficult to conduct over every basin where trends induced by climate change may be
useful to identify. The *d-excess* application introduced in this study can be efficient, applicable across scales that vary
by orders of magnitude, and uses limited labor and instruments for the water sampling (e.g., autosampler) and
standardized laboratory analyses (e.g., laser spectrometer).
The *d-excess* of stream water could serve as a complementary information source in addition to the currently applied
streamflow shape and flashiness at low and high flows to derive relations between snow persistence effects on the
hydrograph across different climates (Le et al., 2022).
Measurements of *d-excess* of stream water could further help disentangle rapid high elevation snowmelt contributions
to the streamflow versus groundwater inflow to the stream. This could be highly beneficial because mountainous
catchments with lower groundwater influence were found to be more sensitive to snowpack changes due to warming
(Tague and Grant, 2009).

## 5 Conclusion

Our snowpack and stream water stable hydrogen and oxygen isotope sampling program during several years links *d-excess* of stream water at the catchment outlet to high elevation snowmelt contributions during the snowmelt period. The relation between *d-excess* of stream water and snowmelt dynamics at high elevations was consistent during several years. End member mixing analyses based on *d-excess* values quantified the temporal dynamics of high elevation snowmelt contributions and its relative importance for the runoff generation from mountainous catchments. As compared to other approaches, such catchment integrated information may be an effective way to better quantify the role of upper subalpine and alpine snowpack for streamflow contributions in snow-dominated mountainous systems. Our findings indicate that high elevation snowpack contributions to the streamflow tend to be more important for runoff generation during years with lower snowpack and warmer spring temperatures. Thus, the high elevation snowpack could likely play a bigger role in the coming decades as snowpack reduces and air temperature rise.

Because we observed an increase of *d-excess* in the stream water during snowmelt for catchments of 85 to over 20,000 km$^2$ in size, the *d-excess* appears to be a robust tracer across a wide range of drainage basin scales. We suggest though that transferability of this approach could depend on the share of high elevation regions of the catchment area that contribute to streamflow, the presence of a *d-excess* lapse rate in the snowpack, and the absence of large reservoirs upstream from the isotope sampling location. With increasing availability of stable isotope data of mountainous catchments across the globe, future synthesis work could investigate the role of high elevation snowmelt contributions in headwater regions worldwide.

## Data availability

The data on East River streamflow (Carroll et al., 2023), snowpack (Carroll et al., 2021), as well as stable isotopes of precipitation, groundwater, and stream water (Williams et al., 2023) are available online as cited. Snow water equivalent data from the SNOTEL sites are made available by NWCC (2023), streamflow and water stable hydrogen and oxygen isotope data from the Gunnison near Gunnison and the Corolardo River near Cameo sites are available from USGS National Water Information System (USGS, 2023) database.

## Code availability

The HydroMix code by Beria et al. (2019) is available on GitHub at https://github.com/harshberia93/HydroMix/tree/20191007_GMD (last access: 20 August 2023).

## Acknowledgements

This work was supported by the US Department of Energy Office of Science under contract DE-AC02-05CH11231 as part of Lawrence Berkeley National Laboratory Watershed Function Science Focus Area. We would like to express appreciation to the Rocky Mountain Biological Laboratory for handling Forest Service permitting. We thank Jarral Ryter in the Western Colorado University Chemistry program for analytical help with Cavity Ring-Down Spectroscopy. Any use of trade, firm, or product names is for descriptive purposes only and does not imply endorsement by the U.S. Government.

**Author contributions**
MS conducted the data analysis and wrote the initial draft of the manuscript. All co-authors contributed either to the
analyses, the database, and the interpretation of both as well as improving the manuscript.
**Competing interests**
The authors declare that they have no conflict of interest.
**Competing interests**
The authors declare no competing interests.

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

*Hydrology and Earth System Sciences*

Supporting Information for

# Stream water sourcing from high elevation snowpack inferred from stable isotopes of water: A novel application of *d-excess* values

Matthias Sprenger[1*], Rosemary W.H. Carroll[2], David Marchetti[3], Carl Bern[4], Wendy Brown[5], Alexander Newman[5], Curtis Beutler[5], Kenneth H. Williams[1,5]

[1]Lawrence Berkeley National Laboratory, Berkeley, CA, USA

[2]Desert Research Institute, Reno, NV, USA

[3]Western Colorado University, Gunnison, CO, USA

[4]U.S. Geological Survey, Denver, CO, USA

[5]Department of Environmental Systems Science, ETH Zurich, Zurich, Switzerland[6]Rocky Mountain Biological Laboratory, Crested Butte, CO, USA

*Corresponding author: msprenger@lbl.gov

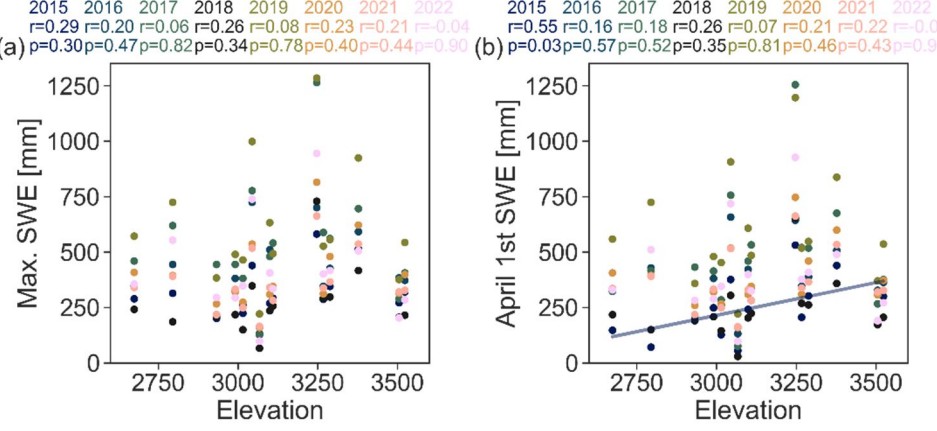

**Suppl. Fig. 1 (a) Relation between maximum snow water equivalent (SWE) at the 15 SNOTEL sites in the Gunnison River basin and the elevation of the SNOTEL sites for the years 2015 to 2022. (b) same as in (a), but with SWE on April 1st. Given are the Pearson correlation coefficients for each year and the years are color coded (data from NWCC, 2023).**

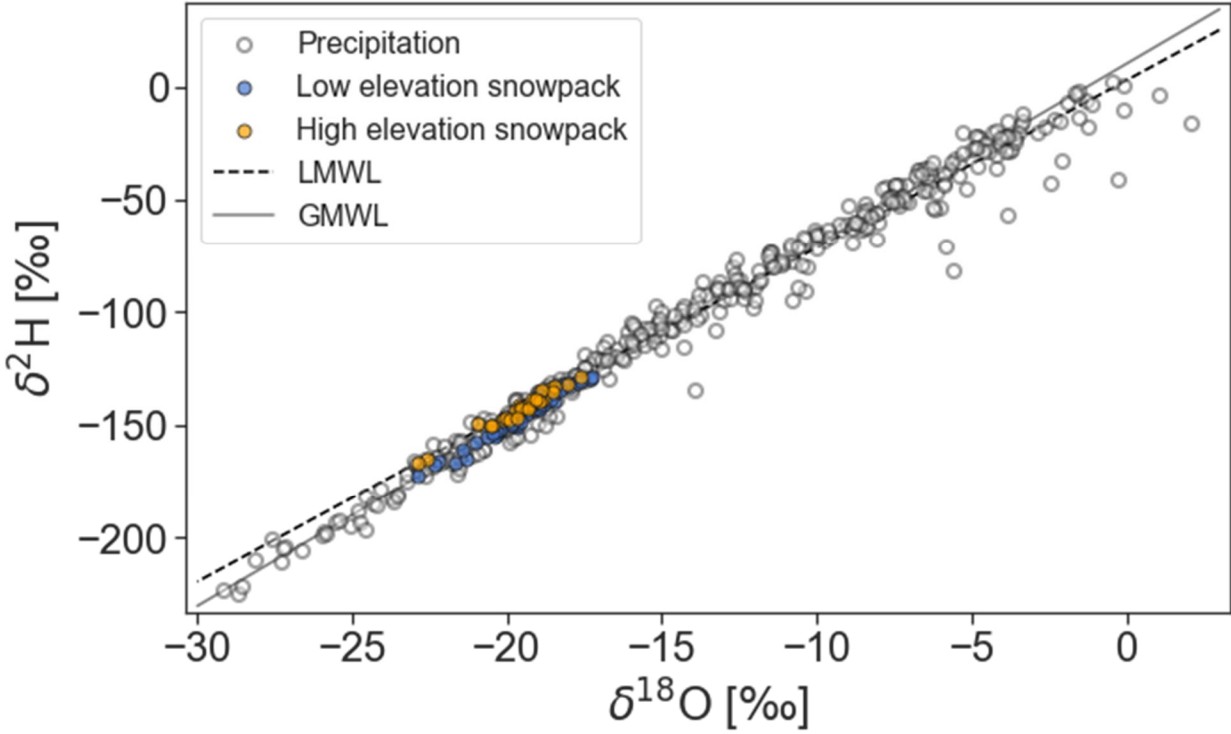

634

**Suppl. Fig. 2 Precipitation samples from 2015 to 2022 (white points) and snowpack sampled at sites <3200 m a.s.l. ("Low elevation", blue) and sites above >3200 m a.s.l. ("High elevation", orange). Also shown are the Global Meteoric Water Line (GMWL: $\delta^2H$ = 8.2 $\delta^{18}O$+11.27, Rozanski et al. (1993)) and the Local Meteoric Water Line (LMWL: $\delta^2H$ = 7.4 $\delta^{18}O$+2.4,Carroll et al. (2022b)).**

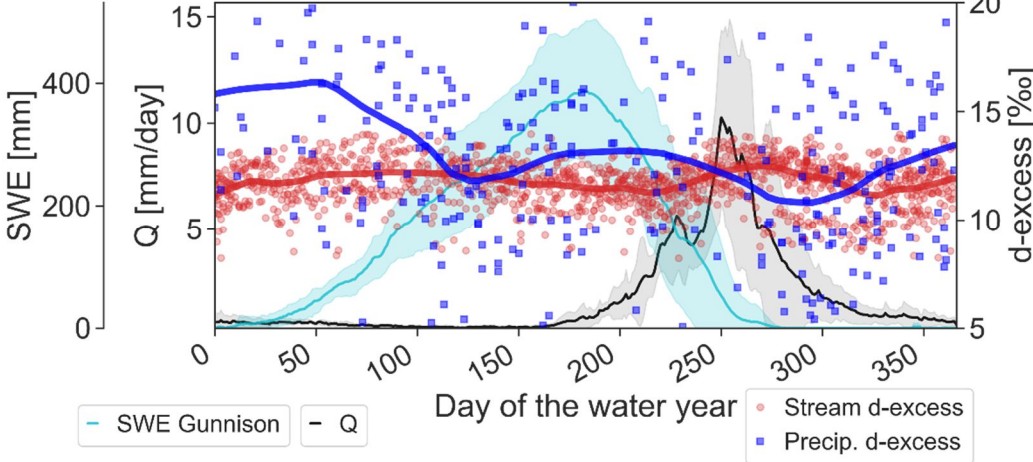

639

**Suppl. Fig. 3 Median annual dynamics of snow water equivalent (SWE) at the Gunnison SNOTEL stations (cyan) and East River streamflow (Q, black) from water year 2015 to 2022 with semitransparent area representing the range. The d-excess of all stream water at East River (red) and precipitation (blue) samples collected between water year 2015 and 2022. The red and blue lines represent a lowess filter to show any trends in the data.**

644

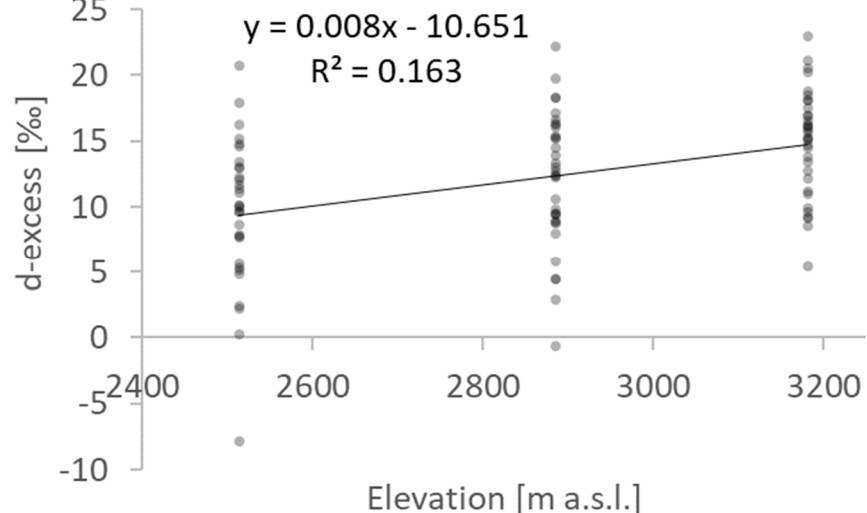

645

**Suppl. Fig. 4 The d-excess of winter precipitation from samples collected between November and April during the water years 2021 and 2022 at the locations Estess (2513 m), Mount Crested Butte (2885 m) and Irwin Barn (3181 m). The black diamonds show the mean values and half-transparent dots are individual samples. The regression line shows the d-excess lapse rate of 0.7 ‰/100 m.**

650

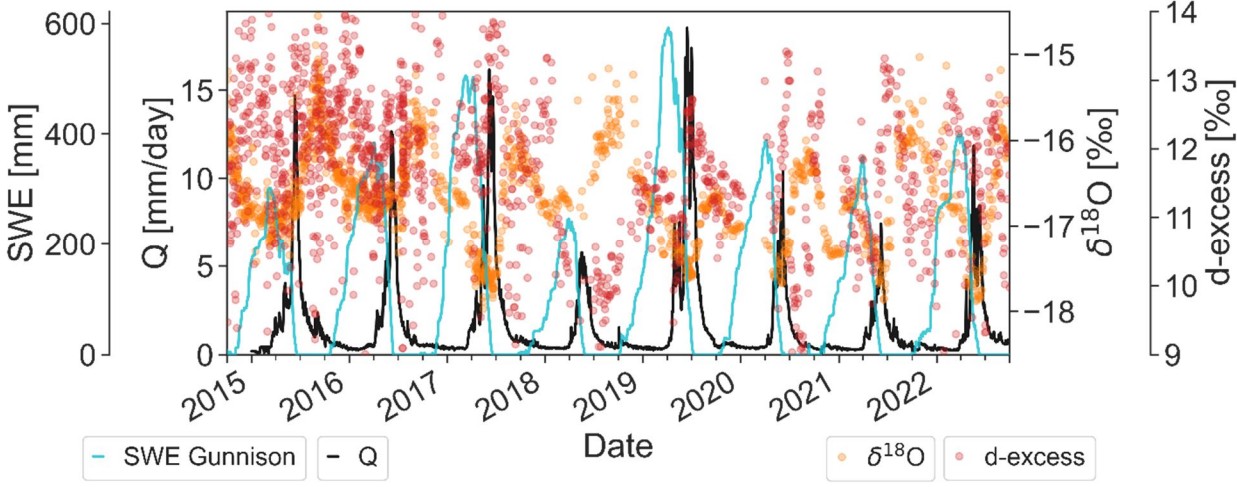

651

**Suppl. Fig. 5 Snow water equivalent (SWE) at the Gunnison SNOTEL stations (cyan line), streamflow (Q, black line) at the East River, as well as the $\delta^{18}O$ (orange points) and *d-excess* (red points) of the stream water sampled at Pumphouse for the water years 2015 to 2022.**

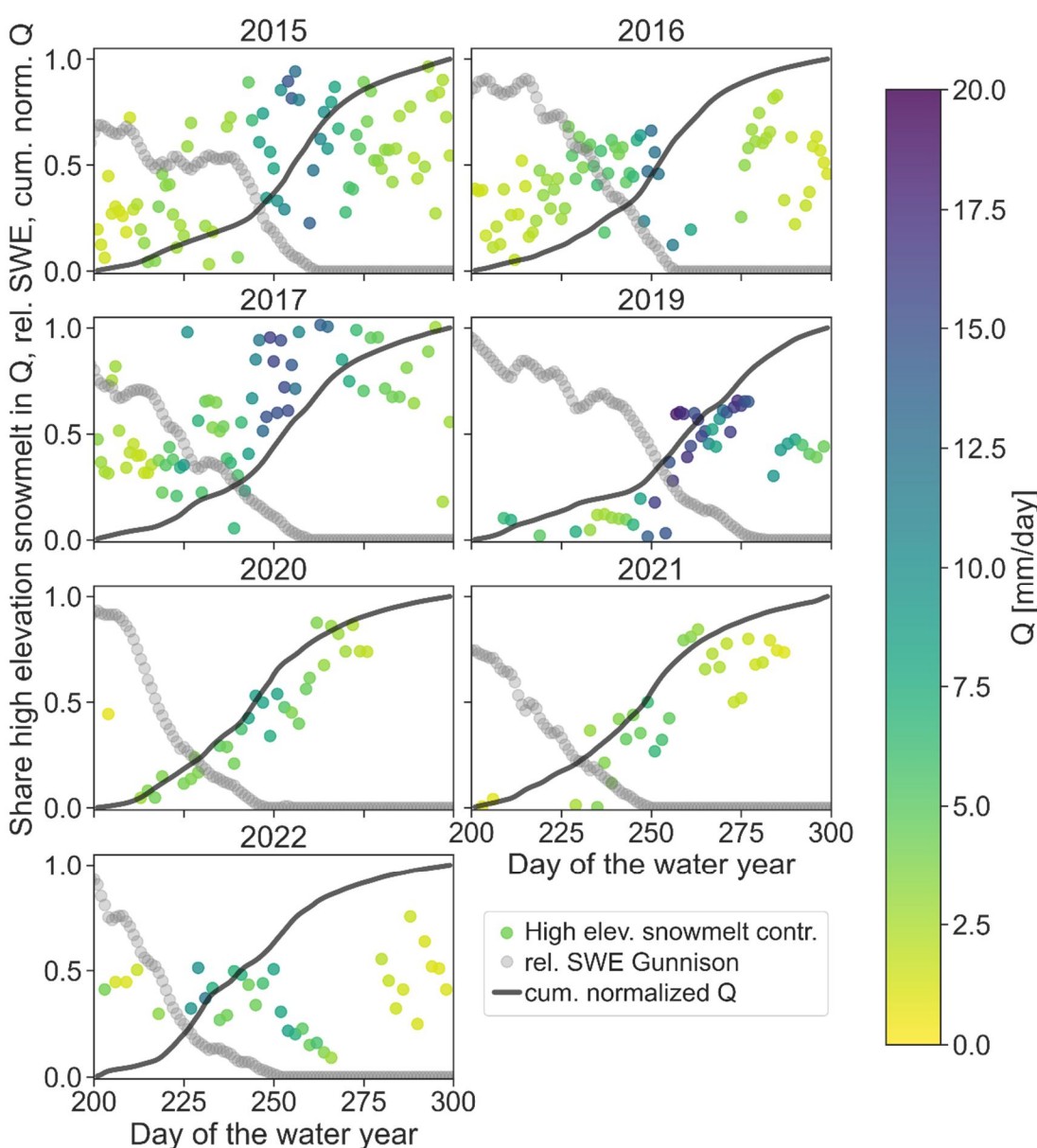

655

**Suppl. Fig. 6 Total streamflow (Q, black line) as well as the snow water equivalent (SWE, cyan) for the SNOTEL sites in**
**the Gunnison catchment. (right) Share of high elevation snowmelt in the streamflow (points, color coded by Q), relative**
**observed SWE in Gunnison (1= maximum SWE), and cumulative streamflow between day 200 and 300 of the water year.**
**Note that the y-axis for the graphs on the right is plotted on the right-hand side.**

660

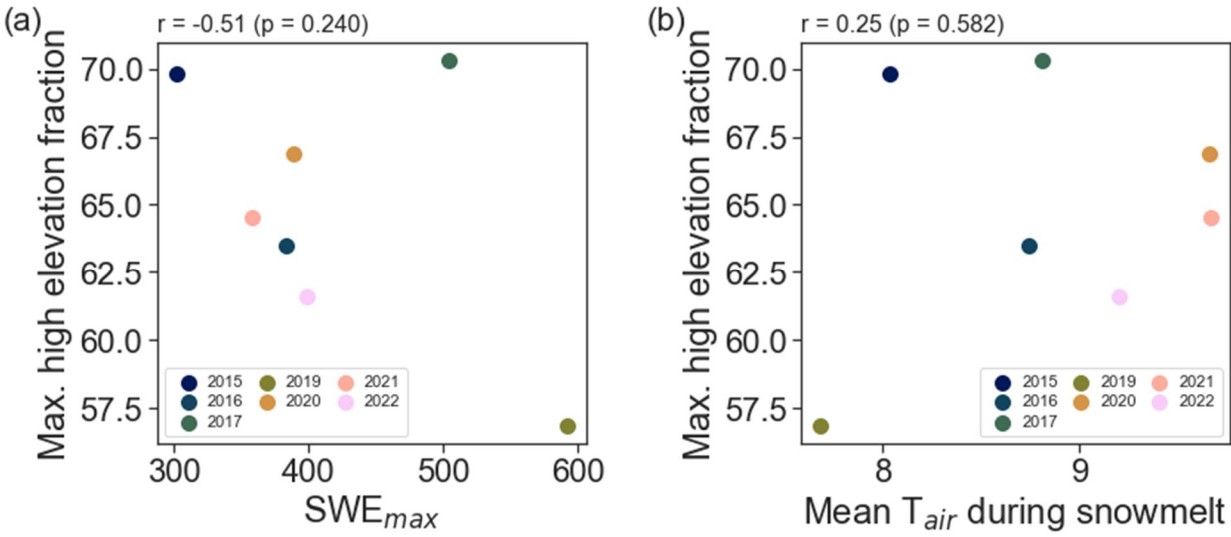

661

**Suppl. Fig. 7 Relation between maximum fraction of high elevation snowpack contributions to the snowmelt runoff and the maximum snow water equivalent (in a) and mean air temperature during the snowmelt period (in b).**

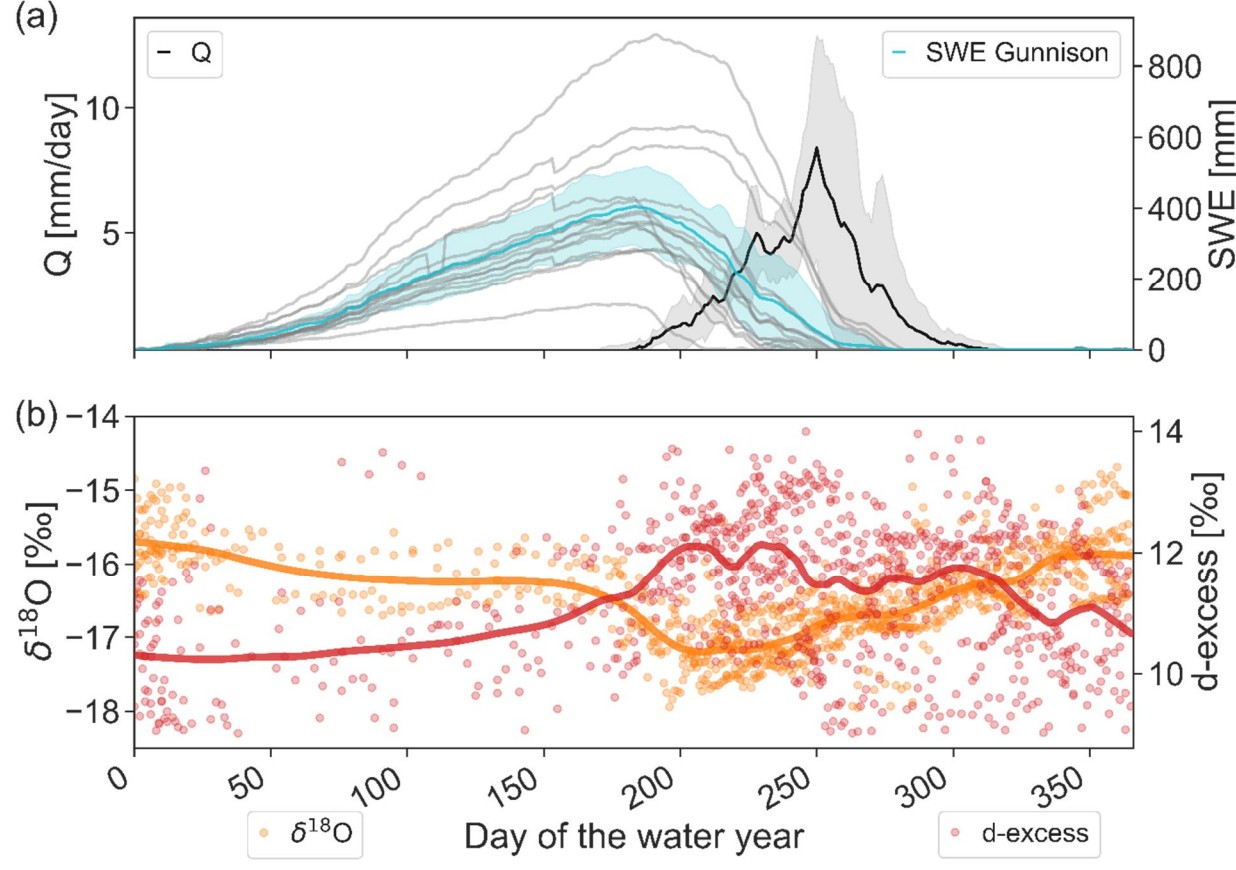

664

**Suppl. Fig. 8 (a) Median annual dynamics of Coal Creek streamflow (Q, black) and snow water equivalent (SWE) at the individual SNOTEL sites within the Gunnison River catchment (grey) and the average of all sites (cyan) from water year 2016 to 2022 with semitransparent grey and cyan area representing the standard deviation of Q and SWE, respectively. (b) The $\delta^{18}O$ (orange) and *d-excess* (red) of all stream water samples collected between water year 2016 and 2022 from the East River at the Pumphouse location. The orange and red lines are a LOWESS fit to the data points.**

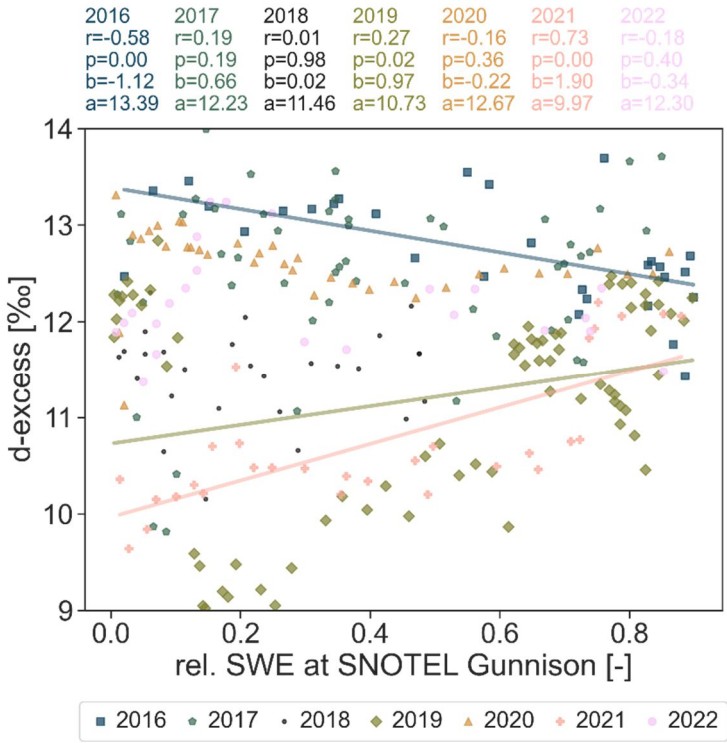

670

**Suppl. Fig. 9 The *d-excess* of Coal Creek stream water during snowmelt for seven individual years, shown as a function of relative SWE measured at the SNOTEL stations across the Gunnison River catchment at the time of sampling. For each year, the Pearson correlation (r) and the associated significance level (p) are given as well as the intercept (a) and slope (b) of the regression.**

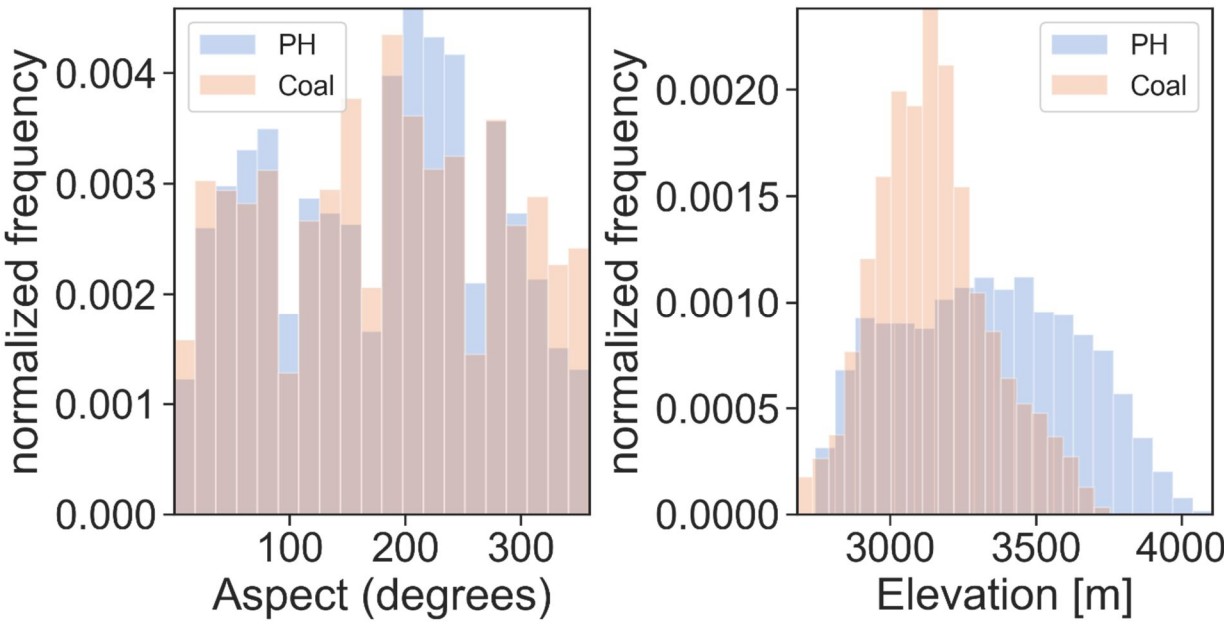


**Suppl. Fig. 10 Distribution of aspect (left) and elevation (right) across the East River catchment defined at Pumphouse (PH, blue) and Coal Creek (Coal, orange).**

**Suppl. Table 1 SNOTEL sites located in the Gunnison River Basin ((data from NWCC, 2023)).**

| Station Id | Station Name | Elevation (m) | Latitude | Longitude | County Name |
|---|---|---|---|---|---|
| 380 | Butte | 3108.96 | 38.8944 | -106.95 | Gunnison |
| 1059 | Cochetopa Pass | 3066.59 | 38.1627 | -106.6 | Saguache |
| 409 | Columbine Pass | 2795.32 | 38.4182 | -108.38 | Montrose |
| 538 | Idarado | 2990.7 | 37.9339 | -107.68 | Ouray |
| 618 | Mc Clure Pass | 2674.32 | 39.129 | -107.29 | Gunnison |
| 622 | Mesa Lakes | 3099.21 | 39.0574 | -108.06 | Mesa |
| 675 | Overland Res. | 3015.39 | 39.0904 | -107.64 | Delta |
| 680 | Park Cone | 2932.48 | 38.8198 | -106.59 | Gunnison |
| 682 | Park Reservoir | 3044.04 | 39.0443 | -107.88 | Delta |
| 701 | Porphyry Creek | 3288.18 | 38.4886 | -106.34 | Gunnison |
| 713 | Red Mountain Pass | 3377.18 | 37.8917 | -107.71 | San Juan |
| 1128 | Sargents Mesa | 3504.9 | 38.2856 | -106.37 | Saguache |
| 737 | Schofield Pass | 3247.03 | 39.0147 | -107.05 | Gunnison |
| 762 | Slumgullion | 3523.49 | 37.9908 | -107.2 | Hinsdale |
| 1141 | Upper Taylor | 3266.54 | 38.9907 | -106.75 | Gunnison |



| Station Id | Station Name | Elevation (m) | Latitude | Longitude | County Name |
|---|---|---|---|---|---|
| 1030 | Arapaho Ridge | 3345.48 | 40.351 | -106.38 | Grand |
| 1061 | Bear River | 2777.34 | 40.0615 | -107.01 | Routt |
| 1041 | Beaver Ck Village | 2610.61 | 39.5987 | -106.51 | Eagle |
| 335 | Berthoud Summit | 3448.51 | 39.8036 | -105.78 | Grand |
| 345 | Bison Lake | 3341.83 | 39.7646 | -107.36 | Garfield |
| 913 | Buffalo Park | 2819.1 | 40.2284 | -106.6 | Grand |
| 1101 | Chapman Tunnel | 3078.48 | 39.2621 | -106.63 | Pitkin |
| 408 | Columbine | 2794.1 | 40.3959 | -106.6 | Jackson |
| 415 | Copper Mountain | 3207.41 | 39.4892 | -106.17 | Summit |
| 1120 | Elliot Ridge | 3215.34 | 39.8638 | -106.42 | Summit |
| 485 | Fremont Pass | 3452.16 | 39.3801 | -106.2 | Summit |
| 505 | Grizzly Peak | 3395.17 | 39.6465 | -105.87 | Summit |
| 542 | Independence Pass | 3230.27 | 39.0754 | -106.61 | Pitkin |
| 547 | Ivanhoe | 3212.9 | 39.2923 | -106.55 | Pitkin |
| 970 | Jones Pass | 3177.84 | 39.7645 | -105.91 | Grand |
| 556 | Kiln | 2933.4 | 39.3172 | -106.62 | Pitkin |
| 565 | Lake Irene | 3255.87 | 40.4145 | -105.82 | Grand |
| 607 | Lynx Pass | 2718.51 | 40.0783 | -106.67 | Routt |
| 618 | Mc Clure Pass | 2674.32 | 39.129 | -107.29 | Gunnison |

| 1040 | Mccoy Park | 2900.48 | 39.6023 | -106.54 | Eagle |
| 622 | Mesa Lakes | 3099.21 | 39.0574 | -108.06 | Mesa |
| 1014 | Middle Fork Camp | 2733.75 | 39.7957 | -106.03 | Grand |
| 658 | Nast Lake | 2661.21 | 39.297 | -106.61 | Pitkin |
| 669 | North Lost Trail | 2809.95 | 39.0782 | -107.14 | Gunnison |
| 675 | Overland Res. | 3015.39 | 39.0904 | -107.64 | Delta |
| 682 | Park Reservoir | 3044.04 | 39.0443 | -107.88 | Delta |
| 688 | Phantom Valley | 2756.92 | 40.398 | -105.85 | Grand |
| 737 | Schofield Pass | 3247.03 | 39.0147 | -107.05 | Gunnison |
| 802 | Summit Ranch | 2856.28 | 39.718 | -106.16 | Summit |
| 842 | Vail Mountain | 3142.49 | 39.6177 | -106.38 | Eagle |
| 869 | Willow Creek Pass | 2902.61 | 40.3473 | -106.1 | Grand |


Any use of trade, firm, or product names is for descriptive purposes only and does not imply endorsement by the U.S.
Government.