# Peer review of "Stream water sourcing from high elevation snowpack inferred from"

_EGUsphere, 2023_

## Author Response (AR1)

**Response to Editor, Markus Hrachowitz**

Dear authors,

as you have seen, the two reviewers have provided very constructive and helpful comments. They both in general appreciate your manuscript and its objectives. However, in particular Reviewer #2 flags a number of issues that need to be addressed. Specifically, more detail is needed in the description of the methods. In addition more importantly, both the Introduction and Discussion Sections will benefit from a wider consideration of potential different processes playing role in the observed pattern and their individual interactions.

I thus encourage you to address these points and the remaining reviewer comments in a round of revisions.

Best regards,

Markus Hrachowitz

*Response:*

*We thank the Editor, Markus Hrachowitz for inviting us to revise our manuscript in accordance to the reviewer comments. We have added to the introduction more about our motivation for the presented study, we have added more information about the applied methods, and we have added to the discussion, potential processes explaining how a d-excess lapse rate in the snowpack develops in mountainous catchments. Several figures were also adjusted as the reviewers suggested. We are grateful for the suggestions provided by the reviewers and believe that the changes in the revised manuscript will both increase the clarity and the impact of our manuscript.*

*We are looking forward to hearing back about the Editor's decision on the revised manuscript.*

**Response to Reviewer #1, James McNamara**

Summary

This is a nice paper proposing that deuterium excess can be used as a tracer to identify stream water sources in snowy catchments. Specifically, they conclude that to proportion of high-elevation snow can be identified in streamflow during snowmelt. This is important since, as the authors state, we typically don't have good measurements of high elevation snow. I think study is worthy of publication in HESS, but I have some suggestions. Below I provide some general comments followed by a few line-by-line comments, followed again by answers to HESS-specific questions.

*Response: We thank James McNamara for taking the time to provide feedback on our manuscript and we appreciate that he agrees that our study is worth publishing in HESS. We provide a response to each of his statements and show how the revised manuscript incorporated suggestions provided by James.*

General Comments

1.       I would like to see a more developed theoretical model. The paper goes from stating some hypotheses in the introduction right to the sampling methods. I think there needs to be a more conceptual approach outlined so that the reader (me) understands why the samples are being collected and how they will be used to test the hypotheses. A paragraph could summarize why d-excess in streams will be different from snow (mixing, fractionation…), assumptions of the mixing model, and then the actual mixing equation. If we see the mixing equation, the sampling methods will make more sense. I understand that the details of the model are in Beria et al (2020), but a general theory here would be helpful.

*Response: We provide the theoretical framework in a revised introduction before we state the hypotheses at the end of the introduction. Additionally, we include more information on the assumptions and background of the mixing model towards the end of section 2.2.*

*We added to the introduction:*

*"Stable hydrogen and oxygen isotopes of water have long been used to infer snowmelt contributions to stream water (e.g., Rodhe, 1981). However, because groundwater recharge is predominantly by snowmelt in snow dominated semi-arid environments (Sprenger et al., 2022), the isotopic difference between snowmelt newly contributing to the stream discharge and the groundwater dominated stream flow during baseflow makes mixing model applications unfeasible in such environments. We therefore explore the applicability of the d-excess value as an alternative tracer."*

*We have added details about the mixing models in the methods section as follows:*

*"We defined the snowmelt period in the East River catchment based on the hydrograph at the Pumphouse streamgage to be the time between day 200 and 300 of the water year. This period is between Mid-April to late July, because the water year starts on October 1$^{st}$. For the snowmelt period, we used the Bayesian mixing model HydroMix, developed by Beria et al. (2020), to estimate the contribution of high elevation snowmelt to streamflow during the snowmelt period. HydroMix uses tracer data of the end-members and the mixture to estimate the probability distribution function (pdf) of the mixing ratio, defined as fractional contribution of end-members to the mixture:*

$$\rho S_1 + (1 - \rho)S_2 = M, \hspace{4cm} (3)$$

*where M is the tracer concentration in the mixture, $S_1$ and $S_2$ are tracer concentrations in the two sources, and $\rho$ is the fractional contribution of $S_1$ to mixture M.*

*In typical Bayesian mixing analysis, pdfs are fitted to tracer concentrations in different end-members and the mixture, and the pdf of the mixing ratio is estimated using standard Bayesian inference principles. This requires a large tracer dataset to ensure a robust fit to tracers of the end-members and the mixture, which is often not available. HydroMix adopts a bootstrap*

*approach, using all possible combinations of end-member tracer measurements and formulating a likelihood function based on an assumed pdf of the underlying error function, which is the difference between simulated and observed mixture concentration. By using all available combinations of end-member tracer measurements, HydroMix builds an empirical pdf while optimizing the likelihood function. This approach has been shown to work both theoretically and in real-case scenarios (Beria et al., 2020).*

*The two end members ($S_1$ and $S_2$) were defined as the d-excess of the snowpack from the upper subalpine and alpine snowpack (>3200 m, n=31, defined as "high elevation") and lower subalpine and montane area (<3200 m, n=60), respectively. We report the mean fraction of high elevation snowmelt in each water sample (M) with standard deviations based on the distribution of the two endmembers as described in Beria et al. (2020). We further report the seasonal flow weighted mean share of high elevation snowpack in the stream samples."*

2.      There are a lot of figures, and it seems arbitrary which ones appear in the main manuscript and which ones are supplemental. Fig 3 and Supp Fig 4 are very similar. Can they be merged? I think Fig sup3 should be in the main document. It's the only place where the reader can actually see that data that goes into the mixing model.

*Response: We moved Suppl. Fig. 3 to the manuscript and added it as a third panel to Figure 2, since these show the same data. The revised Fig. 2 is:*

[Figure]

*However, we believe that Supplementary Figure 4 is not sufficiently important to include it in the manuscript, since it is only provided to show that d-excess values of the stream water do not stem from temporal variability of d-excess values in the precipitation. We would have to add a third*

*sub-plot panel in Figure 3 to include the precipitation d-excess values due to the larger range of d-excess values in precipitation.*

3.      Fig 5 is too complex. Just figuring out which axes go to which plots is challenging. Can you separate the left and right into two figures? If so, can the right figs go into supplemental info? Those panels are really just interpretations of the left panels.

*Response: We simplified Figure 5 as suggested and moved the right panel to the supplementary material.*

*The revised Fig. 5 looks as follows:*

[Figure]

*And the new Supplementary Figure 6 looks as follows:*

[Figure]

4.    Two hypotheses are introduced. Although they are indirectly addressed, the conclusion doesn't specially return to them. If problems are introduced as hypotheses, we need to see them in the conclusion.

*Response: In the introduction we outline the two hypotheses as follows: "First, we hypothesize that d-excess values in stream water during the snowmelt hydrograph reflect the changing dominance of snowmelt contributions through time from lower to higher elevations. Second, we test if these patterns of d-excess of stream water are detectable across ranges in drainage area, thus increasing their broader applicability." With regard to the first hypothesis we pick up this hypothesis in the Conclusion when we write: "The relation between d-excess of stream water and snowmelt dynamics at high elevations was consistent during several years. End member mixing analyses based on d-excess values quantified the temporal dynamics of high elevation snowmelt*

*contributions and its importance for the runoff generation from mountainous catchments."*
*However, we added the results of testing the second hypothesis in a revised Conclusion, as*
*follows.*

*"Because we observed an increase of d-excess in the stream water during snowmelt for*
*catchments of 85 to over 20,000 km2 in size, the d-excess appears to be a robust tracer across a*
*wide range of drainage basin scales. We suggest though that…"*

Line comments

75      Relatively

*Response: We deleted "relative" in the revised manuscript.*

151-154        Why justify not using lc-excess if you don't introduce it anyway?

*Response: The reference to lc-excess was taken out.*

154     "used" instead of "decided to use"

*Response: This part was deleted in response to the previous comment.*

165     Awkward sentence

*Response: This sentence was rewritten in response to the reviewer's request to provide more*
*information about the method.*

376     Pet peeve of mine, but I don't think this is a proper use of hypothesize. Speculate?
Suggest?

*Response: Agreed, we changed this to "suggest".*

Responses HESS Guidelines

1.      Does the paper address relevant scientific questions within the scope of HESS?

Yes, using tracers to identify streamflow sources is a common theme in HESS.

1.      Does the paper present novel concepts, ideas, tools, or data?

Yes. Despite the common theme, d-excess, to my knowledge, has never been proposed as a
tracer.

1.      Are substantial conclusions reached?

Yes, but perhaps unclear. One major conclusion is that their approach is a "cost-effective" way to
quantity snowpack contributions to streamflow. It may be cost effective relative to actual

distributed snowpack observations as described in the introduction, but those methods are used to determine streamflow sources anyway. So, it doesn't make sense to me to state that the method is cost effective since it's doing something different. Is it cost effective relative to other tracer methods?

1. Are the scientific methods and assumptions valid and clearly outlined?

Indirectly. The method uses a new tracer (d-excess) in a previously published application (HydroMix), of standard end-mixing techniques. The novelty here is the use of d-excess. As a reader, I would like to see a summary of end-member mixing theory, followed by a justification of how d-excess meets those assumptions. Since a primary conclusion is about the method, rather than the hydrology, we need to see more method development.

*Response: As mentioned above, we provide more information about the applied method (incl. limitations) in the revised manuscript.*

1. Are the results sufficient to support the interpretations and conclusions?

Yes

1. Is the description of experiments and calculations sufficiently complete and precise to allow their reproduction by fellow scientists (traceability of results)?

No. See point 4. We need to see more method development.

*Response: As mentioned above, we provide more information about the applied method (incl. limitations) in the revised manuscript.*

1. Do the authors give proper credit to related work and clearly indicate their own new/original contribution?

Yes

1. Does the title clearly reflect the contents of the paper?

Yes

1. Does the abstract provide a concise and complete summary?

Yes

1. Is the overall presentation well-structured and clear?

There is A LOT of information and figures in supplemental information. It seems that the choice of which figures went in the main text was somewhat arbitrary. Perhaps the authors could rethink this.

*Response: As mentioned above, we rearranged some of the figures as suggested.*

1.      Is the language fluent and precise?

YEs

1.      Are mathematical formulae, symbols, abbreviations, and units correctly defined and used?

Yes

1.      Should any parts of the paper (text, formulae, figures, tables) be clarified, reduced, combined, or eliminated?

I think the authors should rethink which figs go in the main text vs supplemental, or perhaps reduce the total number of figures.

1.      Are the number and quality of references appropriate?

Yes

1.      Is the amount and quality of supplementary material appropriate?

This is an interesting and relevant paper on a novel use of d-excess for meltwater-source attribution and the fact that patterns of d-excess of stream water are detectable across a wide range of scales is very interesting. The paper is overall well written and presented.

*Response: We thank Bettina Schaefli for taking the time to provide feedback with several suggestions to improve the manuscript. We appreciate that she shares our view that the presented work is interesting and well presented. We provide responses to each of her suggestion and question and we outline how a revised manuscript will address these aspects.*

However, in its present form, it tends, in my view, to be over-optimistic regarding the usefulness of this approach for other hydroclimatic settings. It should probably contain the keyword "semi-arid" already in the title but at the minimum in the abstract and in the conclusion. The paper has relatively few process insights and in particular mentions only once the reason why the method works: because there is a strong d-excess elevational gradient due to sublimation. Such a strong sublimation effect can probably be expected to be visible only in certain climatic settings (in terms of moisture sources, aridity and snow cover seasonality), and only with an appropriate temporal snow sampling strategy (close to peak snow?). Both aspects are not discussed in the paper, which partly limits the reader's insights into potential transferability.

*Response: We extended the discussion on potential limitations of the proposed method in response to the comments to tune down the optimism. However, it is not clear why "semi-arid" would be an important expression in the context of our findings. While the potential evapotranspiration is higher during the growing season than the monsoonal rainfall in the mountainous study region, leading to water limited conditions in parts of the catchment towards the end of the growing season. However, the connection to the d-excess dynamics are not obvious to us.*

The introduction and case study sections do not contain a discussion of the moisture sources for the studied catchments. Is there a dominant moisture source for winter precipitation? Could this play a role in the consistent elevational trend of d-excess or rather: would a strong time variability of moisture sources potentially overrule the sublimation-related d-excess gradient? The d-excess trend is said to be essentially linked to evaporation/sublimation. But why is d-excess lower at lower elevations? Why does sublimation evolve with elevation? Because of differences in radiation? Because snow is more exposed to sublimation at higher elevations (longer snow period)? But: snow at higher elevations is quickly covered by the next snow layer, does sublimation play an effect elsewhere than in the top layer? Is it only sublimation or also processes related to snow metamorphism in the snowpack? In this case, is d-excess simply a measure of how long the snow was sitting on the ground before melting?

*Response: The dominant moisture source in the study region is from the northwest. While we cannot assess with our current sampling design any potential impacts of varying moisture sources, we however mention this aspect in the revised study site description and also pick up that aspect in the revised discussion referring to literature.*

*We have added to the site description: "The dominant moisture source of winter precipitation in the study region is the northeastern Pacific and snowfall occurs predominantly from northwestern frontal storms (Marchetti and Marchetti, 2019)."*

*Regarding the processes leading to an elevation gradient of d-excess was addressed in the revised Discussion section. We focus our revised discussion on fractionation due to lower elevation evaporation and higher deposition/condensation based on by Lambán et al. (2015) and Whal et al. (2021), but also mention fractionation of precipitation due to sub-cloud evaporation processes based on Froehlich et al. (2008), and fractionation due to changes within the snowpack.*

*We added this new paragraph to the discussion:*

*"However, the processes why we see a d-excess lapse rate in mountain snowfall and snowpack is not yet fully understood. The current literature suggests two potential processes:*

*One potential explanation for how d-excess lapse rates in the snowpack develop is evaporation and sublimation of snow at lower elevation combined with daytime up-valley (anabatic) winds that occur in mountainous areas and the subsequent condensation of the water vapor at colder higher elevation (Beria et al., 2018; Lambán et al., 2015). Sublimation and evaporation from the snowpack leads to kinetic non-equilibrium fractionation that leaves an isotopically enriched snowpack behind (Stichler et al., 2001). Recent in situ stable isotope measurements by Wahl et al. (2021) support this process, because they saw that when radiation driven sublimation outweighed deposition, the vapor was isotopically depleted compared to the snowpack. They further showed that the isotopic composition of the vapor determined the isotopic composition of the humidity flux during deposition conditions (Wahl et al., 2021). For our study region, we have shown previously via spatially explicit snowmelt modeling based on the energy balance and accounting for isotopic fractionation* (Carroll et al., 2022) *that the snowpack at lower elevations experience more snow loss to the atmosphere due to higher energy availability than higher elevation, which lead to an elevation gradient of the d-excess in the simulations. These simulations also have shown that shading provided by vegetation in forested areas reduces evaporation and sublimation from the underlying snowpack, making d-excess values of these snowpack higher than snowpack in non-forested areas at the same elevation* (Carroll et al., 2022)*. Because the snowpack in forests with higher d-excess values melt later than the snowpack in non-forested areas, it also results in an increase in stream water d-excess values during the later phase of the snowmelt discharge peak.*

*The second potential explanation for how d-excess lapse rates in the snowpack develop would be sub-cloud evaporation, which leads to lower d-excess values of precipitation at lower elevations, because the distance between cloud base and ground and the saturation deficit are higher than at higher elevations. Thus, precipitation at lower elevations would experience more kinetic non-equilibrium isotopic fractionation due to evaporation leading to lower d-excess (Froehlich et al., 2008). However, this process is less like to occur during winter time and snowfall (Froehlich et al., 2008), and Xing et al. (2023) showed with precipitation and vapor isotope measurements that sub-cloud evaporation altered the d-excess values of snowfall much less than rainfall in the*

*Chinese Loess Plateau. While we cannot conclude which process leads to the d-excess lapse rate, the observation of a d-excess lapse rate in several other high elevation snow studies (Rolle, 2022; Tappa et al., 2016; Froehlich et al., 2008) suggests that we could expect a d-excess response due to high elevation snowmelt contributions in the flow of other mountainous streams. Thus, the transferability of our approach to other watersheds will depend on observations of a d-excess lapse rate in the snowpack, which will likely be influenced by climatic conditions that lead to a thick snowpack without mid-winter melt, relatively steady moisture source of the snowfall, and accessibility to sample the snowpack near peak SWE. "*

What role does the vegetation play (snow interception, snow blowing, sublimation)? Are there mid-winter melt events or what role does the absence of mid-winter melt play for the d-excess gradients? It would be useful to provide these elements to understand how transferable the approach potentially is.

*Response: We have shown via modeling in previous studies that there is no pronounced Mid-winter melt occurring in the subalpine. In the montane region little melt (<10 mm/day) occurs prior to early March (Carroll et al., 2022). We added this information to the study site description as follows: "There is a consistent snowpack cover in the subalpine and alpine region with no mid-winter melt. In the montane region melt is very limited (<10 mm/day) prior to early March (Carroll et al., 2022a)." We also added this aspect that there is no mid-winter melt to the discussion on transferability of the approach.*
*Previous simulation of isotopic fractionation of the snowpack in our study region has shown that evaporation fractionation of the snowpack is lowest in forest and shrub land (Carroll et al., 2022). This was added to the discussion as follows: "For our study region, we have shown previously via spatially explicit snowmelt modeling based on the energy balance and accounting for isotopic fractionation (Carroll et al., 2022a) that the snowpack at lower elevations experience more snow loss to the atmosphere due to higher energy availability than higher elevation, which lead to an elevation gradient of the d-excess in the simulations. These simulations also have shown that shading of vegetation plays a role for the potential of isotopic fractionation of the snowpack, because evaporation fractionation of the snowpack was lowest in forest and shrub land compared to bare land (Carroll et al., 2022a). Because the snowpack in forests will have higher d-excess and melts later compared to the snowpack of lower d-excess in non-forested areas at the same elevation, such a vegetation effect would lead similarly as the lapse rate to an increase in d-excess values in the stream water during the later phase of the snowmelt discharge peak.*

The paper has a very light methods section, which could certainly be enriched with details e.g. on the sampling periods (early or late snow season, with / without fresh snow), on snow modelling (energy balance or degree-day), on the regression analysis and on how the predictors and predictands of the regression analysis are computed from the available data (from station data or modelling?). Why is SWEmax analyzed, is this a good proxy for available snow during a snow season? Why do you predict the maximum share of high-elevation snowmelt contribution to streamflow and not the average? Is the maximum share relevant if streamflow is low?

*Response: We added to the methods section information about the snowpack sampling timing by adding the following sentence: "The snowpack sampling generally took place between early February and late May with 80% of all samples taken +- 30 days of April 1st, which is often assumed to be the timing of peak SWE."*

*We added that we used an energy balance snowmelt model as follows: "…,we used spatially explicit energy balance snowmelt simulations, as published by Carroll et al. (2022a), that were informed by the spatial variation in SWE as observed by flights of the airborne snow observatory (ASO)."*

*As written in the methods section, the regression analysis was done with SNOTEL data. We revised the section as follows to clarify how the predictors were computed: "Multiple linear regression was used to explore the predictability of the mean share of high elevation snowmelt during the different years as a function of the average maximum SWE (SWEMax) and the mean air temperature (Tair) of measurements at the Gunnison SNOTEL sites during the snowmelt period."*

*We present in the revised manuscript the mean high elevation snowmelt contribution. In fact, the presented multiple linear regression model showed the mean instead of the max, but the manuscript text and legend in Fig. 6 wrongly showed max instead of mean in the initial submission.*

Another general comment is about the considered elevational zones: How were the limits identified? Do they have any particular snow-hydrological relevance (see also below)?

*Response: The zones were based on the dominant vegetation in the elevation zones. We clarified this as follows in the revised manuscript: "Varying dominance of vegetation with elevation define four ecozones in the catchment: shrubs, grasses, and forbs dominate the montane (<2800 m elevation, 2% of catchment area) zone, aspen and conifers dominate in the lower subalpine (2800 to 3200 m, 34% of the catchment area) region, and conifers dominate in the upper subalpine (3200 to 3500 m, 32% of the catchment area) region. In the alpine region (>3500 m, 31% of the catchment area), shrubs are dominant until 3800 m, above which land is mostly barren (Carroll, Deems, Sprenger, et al., 2022)."*

Finally: part of the reasoning is based on "low-snow years"; the text seems to stipulate that in low-snow years, there is little snow in all elevation zones, versus in high snow years, there is a deep snowpack everywhere. This is probably the case in this catchment, but: we do not know what causes low-snow years; below-average precipitation? or above-average air temperature (thus little snowfall)? or mid-winter melt events at lower elevations? Or a mixture of everything? Can we have years with high snowpacks at high elevations but low snow packs (temperature-limited, later onset) at low elevations? If yes, perhaps the reasoning in terms of low versus high snow years is not sufficient.

*Response: Low-snow years in the study region is mainly due to a lack of winter precipitation. As mentioned above, mid-winter snowmelt is not prevalent in this catchment. The maximum SWE of the SNOTEL sites at elevation <3200m was linearly correlated with the maximum SWE of the SNOTEL sites at elevation >3200m (r=0.97, p<0.0001). However, lower elevation SNOTEL sites*

*reported maximum SWE that were on average 80% of the maximum SWE of the high elevation SNOTEL sites. Temperature is also an important factor in addition to the maximum SWE because we can have relatively warm snowmelt period despite a thick snowpack (e.g., 2017), which leads to mean high elevation snowmelt contributions that are higher than one would expect from the maximum SWE alone. In this case, the melt process will be faster leading to a higher snowmelt contribution from high elevations to the snowmelt peak hydrograph. We added this aspect to the discussion as follows:*

*"The observed regression stems from the generally higher volume share of high elevation snowpack compared to low elevation snowpack during low snow years, and the faster melt out during warmer spring temperatures, both leading to larger contributions of high elevation snowmelt to the spring hydrograph peak."*

Detailed comments:

- I think there is no general agreement in hydrology on what is subalpine, alpine, high-elevation etc.; however, as far as I understand, there is an ecological agreement on "alpine" meaning above the treeline; I would thus give details on where the classification into subalpine, upper subalpine, montane and alpine (being here > 3500 m asl) comes from; what are the relevant elevations and why are these zones relevant for the case study catchment What is the maximum elevation? Why are the chosen elevational areas relevant? From a general hydrology perspective, we are first of all interested in elevations with intermittent snow versus with seasonal snow cover or interannual snow cover. Furthermore, we might be interested in elevations that receive rain-on-snow versus those that don't. What limit does the 3500 m of the paper correspond to in the studied region?

*Response: We revised the part of the methods section that introduces the elevation zones as follows: "Varying dominance of vegetation with elevation define four ecozones in the catchment: shrubs, grasses, and forbs dominate the montane (<2800 m elevation, 2% of catchment area) zone, aspen and conifers dominate in the lower subalpine (2800 to 3200 m, 34% of the catchment area) and conifers in the upper subalpine (3200 to 3500 m, 32% of the catchment area) regions. In the alpine region (>3500 m, 31% of the catchment area), shrubs are dominant until 3800 m, above which land is mostly barren (Carroll, Deems, Sprenger, et al., 2022)."*
*We further clarified in the methods section that there is no mid-winter melt: "There is a consistent snowpack cover in the subalpine and alpine region with no mid-winter melt. In the montane region melt is very limited (<10 mm/day) prior to early March (Carroll et al., 2022a)."*
*The maximum elevation of the catchment is given in the section 2.1 and a distribution of the elevation is given in the Suppl. Fig. 10 of the originally submitted publication.*

- Sampling: did you mostly sample close to peak SWE? Do you have samples shortly after snowfall events (including fresh snow)?

*Response: We added to the methods: "The snowpack sampling generally took place between early February and late May with 80% of all samples taken +- 30 days of April 1st, which is often*

*assumed to be the timing of peak SWE." As noted in the methods section, we sampled bulk snowpack isotopic content that represents the SWE-weighted composite value across the entire snow column. The potential impact of fresh snow is therefore limited and is not considered in the sampling nor analyses.*

- It is stated: "for snowpacks, the d-excess values were found to increase with elevation (Froehlich et al., 2008; Tappa et al., 2016) due to increased evaporative fractionation from lower elevation snowpacks which are re-condensed at higher elevations (Lambán et al., 2015)." Where (in what climate) has this been found? Where can we assume to hold it (we do not see it in our Swiss samples)? Does it hold if you sample towards the peak of the snow accumulation season or regardless of timing? And what is meant by "which are re-condensed at higher elevations", what is "which" referring to here?

*Response: We realized that this comment, in section 2.2., regarding other studies did not come up at the correct part of the manuscript. We therefore left it out in 2.2. in the revised manuscript. Instead, we added this information to the discussion section. We rewrote that part of the discussion as follows to provide more information on these cited studies. We added a new study by Rolle (2022) that also sampled snowpack along an elevation gradient:*
*"Other studies have also shown that winter precipitation (i.e., snow) at highest elevations had the highest d-excess values; monthly weighted precipitation data by Froehlich et al. (2008) indicated a lapse rate in d-excess values of +0.2 ‰/100 m across an elevation range between 469 and 2245 m in the Alps. Data published by Tappa et al. (2016) indicated a lapse rate of +0.63 ‰/100 m in the Rocky Mountains in Idaho for samples taken between October and May across five sites spanning an elevation gradient from 830 to 1850 m. Rolle (2022) sampled the snowpack at ten sites across elevations from 1262 and 1905 m in the Lubrecht Experimental Forest, Greenough, MT in late March and found a d-excess lapse rate of +0.26 ‰/100 m. Our lapse rate of +0.72 ‰/100 m for precipitation and +0.52 ‰/100 m for the snowpack was higher than in the other studies, but we cover a larger elevation gradient and study higher elevations than the other studies. Nevertheless, the general trend of increased d-excess values with elevation was the same for all four studies in mountainous systems."*

- Manuscript text: "The 18O of snowmelt stream water reached a minimum in June during maximum snowmelt contribution, after which the snowpack ceased to exist and 18O of stream water increased throughout the summer with recession to base flow and monsoonal rainfall." The timing of snowmelt peak in Fig. 3 seems delayed with respect to the SWE (which is already very low at that moment in time), could this be due to the fact that SNOTEL data is not representative for the entire catchment? Is maximum streamflow related to the moment when snow disappears or to the moment when the snowpack has an optimal spatial coverage to provide a maximum of melt?

*Response: As described in the methods section, there are "15 SNOTEL sites located at elevations ranging between 2674 and 3523 m". Thus, the highest elevations in the study region are not accounted for in the SNOTEL SWE data. We did some additional analyses and saw that neither the average timing of peak SWE, nor the average of the higher (>3200m) or the lower (<3200m) elevation SNOTEL stations are a good predictor for the timing of peak Q. However, the timing of both the most intense snowmelt and the complete melt out at the higher (>3200m) SNOTEL stations explain the timing of the peak Q (r=0.83 and r=0.79, respectively).*
*We included these findings in the results as follows:*

*"We found that over the study period, the timing of the peak stream flow could be explained by the timing of the most intense snowmelt (i.e., slope of SWE in Figure 3) and the timing of the complete melt out at the higher (>3200 m) SNOTEL stations (r=0.83 and r=0.79, respectively)."*

*We further added to the discussion section to address the processes involved: Much of the high elevation snowmelt has relatively short travel times, because the highest snowmelt streamflow can be related to high elevation snowmelt based on the endmember mixing model and the high d-excess values observed in high elevation snowpack is not found in the groundwater. Thus groundwater, which has 18O and 2H isotope ratios that are similar to snowmelt, is mainly recharged from lower elevation snowmelt, because the d-excess is about 10‰. We have added the following: "Since the d-excess values in the groundwater are more similar to the lower elevation snowpack (Figure 2c), we infer that groundwater recharge is dominated by early snowmelt in relatively lower elevations infiltrating into a relatively dry subsurface. High elevation snowmelt occurs during later freshet when the soils are already saturated or near saturation, which leads to fast runoff generation and thus shorter travel times and higher runoff efficiency of high elevation snowmelt than low elevation snowmelt."*

- Does the snowmelt transit through the groundwater (which would have deltaO18 /d-excess values close to snowfall)? and could this explain the delay between the minimum delta O18 / d-excess values and maximum peak flow?

*Response: As mentioned in the previous response, it seems that the earlier snowmelt is dominating the groundwater recharge, while much of the later, high elevation snowmelt is routed in relatively shallow flow paths. We added these processes to the discussion section*

- Manuscript text: "Instead, d-excess of stream water resulted from melting snowpack at higher elevations due to snowmelt progression, as evidenced by the SNOTEL SWE data, that resulted in increases in d-excess of stream water consistently for each of the investigated years (Figure 4a)." I understood from the earlier parts of the paper that SNOTEL data does not show a consistent elevational trend in terms of SWE dynamics (line 190) and that SNOTEL it is not representative for actual snow conditions in the catchment. It is perhaps not optimal to then invoke here the relationship between d-excess and SNOTEL data.

*Response: We use the SNOTEL SWE as one line of support among others to show the process behind the observed d-excess value changes in the streamflow during the snowmelt hydrograph*

*peak. Only a few catchment studies have airborne snowpack observations (ASO) available like we do, but SNOTEL sites are covering large areas of the snow impacted Western US. Additionally, SNOTEL data are continuous measurements, while ASO data are limited to snapshots. We therefore use SNOTEL data in our analyses to relate our observations to more commonly available data sets.*

- Manuscript text: "When the high elevation snowmelt volumes became increasingly larger than the low elevation snowmelt, d-excess of stream water increased consistently." Just for additional information: how large are the "montane" areas and the high elevation areas"? Since the mm of melt in Figure 4b are relative to the entire catchment area, it would be nice to also know how large snowmelt gets at high elevations (relative to the area of the high elevation). This information is missing.

*Response: The simulations were done for a 750km2 catchment area of the East River and the Montane region was 143 km2 in size and the Alpine region was 111 km2 in size. Thus, the Montane region was larger in size, but there was on average double as much snowmelt per year from alpine regions (1075 m3/s) than from montane regions (520 m3/s).*
*We included this information about the absolute volumes in the revised manuscript as follows: "Annual average snowmelt from alpine regions (1075 m3/s) was more than double than snowmelt from montane regions (520 m3/s), despite the area of the prior (111 km2) being smaller than the latter (143 km2) in Carroll (2022a)'s modeling domain of the East River."*

- Manuscript text: "Because of this observation, we included the average air temperature measured at the SNOTEL sites during the snowmelt period as a second variable in a multiple regression analysis." How did you determine the snowmelt period (should be in the methods)? What was the first variable, peak SWE? If yes, peak SWE at a station or from the model? The information is probably in the SuppMat. What is the predictand? Becomes clear only in the figure 6 caption that it is "maximum high elevation fraction". Why is the maximum fraction relevant rather than the average fraction (more interesting from a water resources perspective?).

*Response: We added the following definition to the methods section: "We defined the snowmelt period in the East River catchment based on the hydrograph at the Pumphouse streamgage to be the time between day 200 and 300 of the water year. This period is between Mid-April to late July, because the water year starts on October 1st."*

*At the end of the methods section we describe the multiple linear regression. The predictors are observations from the SNOTEL station. We actually had calculated the mean of the high elevation snowpack contributions, but the plot and text said maximum high elevation contribution. This was corrected in the revised manuscript. We added the variable names of the two predictors and changed the section in response to the comment as follows: "Multiple linear regression was used to explore the predictability of the mean share of high elevation snowmelt during the different years as a function of the average maximum SWE ($SWE_{Max}$) and the mean air temperature ($T_{air}$) of measurements at the Gunnison SNOTEL sites during the snowmelt period." We will further*

*rephrase the results section as follows so that it is clear what the predictors and the response variable are – including the regression equation: "There was a general trend that the annual mean high elevation snowpack contributions were higher in water years with lower maximum SWE observed at the SNOTEL sites across Gunnison county (Suppl. Fig. 7a, $\rho$=-0.39, p=0.383). However, the relatively warm snowmelt period of 2017, following a winter with deep snowpack, resulted in relatively large high elevation snowmelt contributions and thus did not follow that trend. Because of this observation, we included in addition to maximum SWE also the average air temperature measured at the SNOTEL sites during the snowmelt period as a second variable in a multiple regression analysis. The regression mean high elevation snowmelt contribution = -37.03\*$T_{air}$ - 0.73\*$SWE_{max}$ + 0.089\*$T_{air}$\*$SWE_{Max}$ + 350.74 explained 66% of the interannual variation of the mean high elevation snowmelt contribution, and all variables had significance levels of <0.1. Our results therefore indicate that the snowpack at the highest elevation can be more important for runoff generation in low-snow years and when the air temperature is higher (Figure 6). We also tested the streamflow volumes during the snowmelt period as a variable, but did not include it, because of its strong correlation with $SWE_{max}$ (r=0.84, p=0.018)."*

- Fig. 5: can you add that day 300 is the end of July? left: how is the single SWE line obtained, by averaging all station data or by averaging selected stations or from a single station?; right: is the relative SWE from data or modelling?

*Response: We simplified this figure in accordance to Reviewer #1's comment. We show the left part and moved the right part to the supplementary material. We added the requested information as follows to the figure's caption: "Days 200 and 300 of the water year represent Mid-April and late July, respectively. The cyan line represents the average snow water equivalent (SWE) observed across the SNOTEL sites in Gunnison county."*

- Fig. 5: right: what explains the high share of high-elevation water at low flow in July (yellow dots at the right side of the plot)? Some leftover snow at high elevations that continues to melt while there is no other input?

*Response: We can see based on the mixing model results that the relative share of high elevation snowmelt tends to increase towards the later part of the snowmelt peak. The reason for this is the earlier melt out in lower elevations and later melt at higher elevations as also mentioned by the reviewer. We clarified this interpretation in the discussion section as follows:*

*"Since the d-excess values in the groundwater are more similar to the lower elevation snowpack (Figure 2c), we infer that groundwater recharge is dominated by early snowmelt in relatively lower elevations infiltrating into a relatively dry subsurface. High elevation snowmelt occurs during later freshet when the soils are already saturated or near saturation, which leads to fast runoff generation and thus shorter travel times and higher runoff efficiency (as outlined by Webb et al., 2022) of high elevation snowmelt than low elevation snowmelt. This temporal aspect of the high elevation snowmelt and its larger contribution to streamflow later in the snowmelt hydrograph is reflected in the endmember mixing results that show the highest share on the recession limb of the hydrograph."*

- Fig. 6: what does the figure show, the prediction of the regression? here and elsewhere; how do you obtain SWEmax from SNOTEL sites? Perhaps the methods section should mention this; the regression equation could be in the text instead of in the figure caption (as it is now, we do not even know what the predictand is).

*Response: As mentioned above, in the revised manuscript we show the resulting regression equation in the manuscript text and clarify that the predicted variable was the mean high elevation snowmelt contribution to the stream flow. We clarified in the methods section that $SWE_{max}$ was derived from the SNOTEL observations.*

- Fig. 6: the comment on this figure in the text is "This regression explained 66% of the interannual variation of the maximum high elevation snowmelt contribution, and all variables had significance levels of <0.1. Our results therefore indicate that the snowpack at the highest elevation can be more important for runoff generation in low-snow years and when the air temperature is higher (Figure 6)" The figure shows that high fractions appear for low SWEmax but high temperatures or high SWEmax and low temperatures (upper left corner), both are to be expected, but not both commented on in the text.

*Response: We extended the discussion of the results from the regression analyses. We further added to the Figure 6 the data points of the seven years that went into the model.*

*In the results section we rephrased this part as follows:*

*"The regression equation*
*mean high elevation snowmelt contribution = -37.03\*Tair -0.73\*SWEmax + 0.089\*Tair\*SWEMax + 350.74 (4)*
*explained 66% of the interannual variation of the mean high elevation snowmelt contribution, and all variables had significance levels of <0.1. Our results therefore indicate that the snowpack at the highest elevation was most important for runoff generation in low-snow years and relatively high air temperature and years with a deep snowpack and relatively low air temperature (Error! Reference source not found.)."*

[Figure]

- "However, the sampling strategies for the different studies are different, and importantly, the general trend of increased d-excess values with elevation was the same for all three studies in mountainous systems." Sentence is hard to understand. We do not know what the sampling strategy of the present manuscript was; what was the temporal sampling strategy of the other published studies you are referring to? We do not see the same d-excess trend in our own data from the Alps, so it would be really interesting to know what the different sampling strategies were with respect to timing (not just with respect to sampling the snow column).

Response: We added to the methods section the following sentence to clarify the sampling strategy: *"The snowpack sampling generally took place between early February and late May with 80% of all samples taken +- 30 days of April 1st, which is often assumed to be the timing of peak SWE." We further added more information about the sampling and location of the studies we refer to and the sentence referred to in the comment was deleted during the revision. The new revised paragraph comparing the other studies with our results was shown in a response above.*

- Manuscript text: "A potential explanation for how d-excess lapse rates in the snowpack develop is sublimation of snow at lower elevation and the subsequent condensation of the water vapor at colder higher elevation." Does this mean condensation of the same water vapor? Why does the water vapor travel upstream and condensate there on the snowpack, did someone show this? Or did I misunderstand the sentence?

*Response: We addressed this question in the response to the general comments further above (i.e., reference to processes described in Lambán et al. (2015) and Whal et al. (2021))*

- Manuscript text: "We hypothesize that transferability of this approach could depend on the share of high elevation regions of the catchment area to contribute to streamflow, the presence of a d-excess lapse rate in the snowpack, and the absence of large reservoirs upstream from the isotope sampling location." It would be great if you could comment on how transferability depends on i) the temporal variability of moisture sources (could elevational trends be wiped out by differences related to moisture sources?), ii) the seasonality of the snow packs at the considered elevational ranges (with or without winter melt), iii) the sampling strategy (timing relative to peak snow). It would be interesting to discuss how the results depend on snowmelt sampling timing, e.g. on how close to the snow accumulation peak you get the samples. Sampling at peak SWE might be challenging in many locations due to avalanche risk.

*Response: As outlined in more detail in several responses to comments further above, we included more information on the sampling strategy in the methods section and included a more detailed assessment of the limitations of the proposed application of d-excess as a tracer for high elevation snowmelt addressing the reviewer's suggestions.*

Response to Community Comment

This is a really cool study. The data being collected and publications coming out of the East River have been providing some great insights towards hydrologic processes. This study also reminds me of a number of studies on Niwot Ridge that actually find similar results across an elevational gradient that further support the idea of this method being broadly applicable. The studies use slightly different tracers, but also deuterium, and generally look at storage and/or flow paths. However, I think that since Niwot Ridge is also in the CO Rockies they provides further evidence for the processes being discussed and tracers being applied to be used for the CO River Basin. I wonder if the East River is showing similar storage and release mechanisms as what has been observed on Niwot to give insights towards CO River Basin Models. Any thoughts on that given all the data in the East River and comparisons with the studies below?

Sorry to provide so many references below, but I thought the similarities were quite surprising and shows the broad applicability of your study.

Cowie et al., 2017 showed a similar elevational gradient:
https://www.sciencedirect.com/science/article/pii/S0022169417301907

Recently, Webb et al., 2022 and 2018 has also discussed specific flow paths and hydrologic connectivity with elevation that may be of interest.

https://onlinelibrary.wiley.com/doi/full/10.1002/hyp.14541

https://onlinelibrary.wiley.com/doi/full/10.1002/hyp.13686

Lastly (but certainly not least) Liu et al, 2004 and Williams et al., 2015 discuss the specific storage mechanisms in high alpine environments.

https://agupubs.onlinelibrary.wiley.com/doi/10.1029/2004WR003076

https://www.tandfonline.com/doi/full/10.1080/17550874.2015.1123318

*Response:*

*We thank Ryan Webb for the interest in the work coming out of the East River in general and we appreciate the feedback provided for this particular manuscript in currently review for HESS. The studies on Niwot Ridge that he refers to are of general relevance for the hydrological work conducted at the East River.*

*We discuss below each of the manuscript that Ryan Webb referred to and briefly outline its relevance for and potential changes of a revised manuscript.*

*Cowie et al. (2017) showed based on EMMA how groundwater contributions to stream flow decreases with decreasing elevation at Niwot Ridge. Our mixing analyses based on d-excess values cannot identify groundwater as an explicit end-member, because groundwater, rainfall, and lower elevation snowmelt all have very similar d-excess values (about 10 per mill). Carroll et al. (2018) did not find that the fraction of groundwater increased with lower elevation of the catchments, because they reported that fraction of groundwater correlated with snow water equivalent, which correlates with elevation. Catchments with lower snow water equivalent were at lower elevation and had lower groundwater contributions compared to catchments with higher snow water equivalent at higher elevation.*

*If the comment that "Cowie showed a similar elevational gradient" refers to the change of groundwater contributions to stream flow with elevation, there is very limited connection to our current study.*

*The comment refers to Webb 2018 and 2020, but the links provided lead to a paper from 2020 and 2022, respectively. Webb et al. (2020) showed that at higher elevation there was intra-snowpack water flow, whereas at lower elevation such a process decreased and ceased at the lowest observed elevation. Webb et al. (2022) outlines a more comprehensive conceptual model of the runoff generation processes in snow covered slopes.*

*Liu et al. (2004) and Williams et al. (2015) are as well focusing on runoff mechanism highlighting that at high elevations much of the snowmelt is flowing through the relatively shallow subsurface (talus in their catchments). We will include the findings from the Niwot Ridge in our discussion to provide potential explanations how a relatively large share of the high elevation snowmelt is ending up in the streamflow within several weeks during the snowmelt peak.*

*We have included Webb et al. (2022) in the discussion section where we outline the snowmelt process at high elevations leading to faster runoff when soils are already saturated later in the snowmelt period.*

*References:*

Beria, H., Larsen, J. R., Ceperley, N. C., Michelon, A., Vennemann, T., and Schaefli, B.: Understanding snow hydrological processes through the lens of stable water isotopes, Wiley Interdiscip. Rev. Water, 5, e1311, https://doi.org/10.1002/wat2.1311, 2018.

Beria, H., Larsen, J. R., Michelon, A., Ceperley, N. C., and Schaefli, B.: HydroMix v1.0: a new Bayesian mixing framework for attributing uncertain hydrological sources, Geosci. Model Dev., 13, 2433–2450, https://doi.org/10.5194/gmd-13-2433-2020, 2020.

Carroll, R. W. H., Deems, J., Sprenger, M., Maxwell, R., Brown, W., Newman, A., Beutler, C., and Williams, K. H.: Modeling Snow Dynamics and Stable Water Isotopes Across Mountain Landscapes, Geophys. Res. Lett., 49, e2022GL098780, https://doi.org/10.1029/2022GL098780, 2022.

Froehlich, K., Kralik, M., Papesch, W., Rank, D., Scheifinger, H., and Stichler, W.: Deuterium excess in precipitation of Alpine regions - Moisture recycling, Isotopes Environ. Health Stud., 44, 61–70, https://doi.org/10.1080/10256010801887208, 2008.

Lambán, L. J., Jódar, J., Custodio, E., Soler, A., Sapriza, G., and Soto, R.: Isotopic and hydrogeochemical characterization of high-altitude karst aquifers in complex geological settings. The Ordesa and Monte Perdido National Park (Northern Spain) case study, Sci. Total Environ., 506–507, 466–479, https://doi.org/10.1016/j.scitotenv.2014.11.030, 2015.

Rolle, J.: Determining Spatial Controls on Snow Isotopic Signature and Tracing the Snowmelt Pulse as it Moves Through Two Montane Tracing the Snowmelt Pulse as it Moves Through Two Montane Catchments Catchments, Graduate Student Thesis, The University Of Montana, 2022.

Stichler, W., Schotterer, U., Fröhlich, K., Ginot, P., Kull, C., Gäggeler, H., and Pouyaud, B.: Influence of sublimation on stable isotope records recovered from high-altitude glaciers in the tropical Andes, J. Geophys. Res. Atmospheres, 106, 22613–22620, https://doi.org/10.1029/2001JD900179, 2001.

Tappa, D. J., Kohn, M. J., McNamara, J. P., Benner, S. G., and Flores, A. N.: Isotopic composition of precipitation in a topographically steep, seasonally snow-dominated watershed and implications of variations from the Global Meteoric Water Line, Hydrol. Process., 30, 4582–4592, https://doi.org/10.1002/hyp.10940, 2016.

Wahl, S., Steen-Larsen, H. C., Reuder, J., and Hörhold, M.: Quantifying the Stable Water Isotopologue Exchange Between the Snow Surface and Lower Atmosphere by Direct Flux

Measurements, J. Geophys. Res. Atmospheres, 126, 1–24,
https://doi.org/10.1029/2020JD034400, 2021.

Xing, M., Liu, W., Hu, J., and Wang, Z.: A set of methods to evaluate the below-cloud evaporation
effect on local precipitation isotopic composition: a case study for Xi'an, China, Atmospheric
Chem. Phys., 23, 9123–9136, https://doi.org/10.5194/acp-23-9123-2023, 2023.